# PAWH1 and PAWH2 are plant-specific components of an Arabidopsis endoplasmic reticulum-associated degradation complex

Liangguang Lin [1,2,7], Congcong Zhang[1,2,7], Yongwu Chen[1,2], Yi Wang[1,2], Dinghe Wang[1,3], Xiaolei Liu[1], Muyang Wang [3], Juan Mao [4,5], Jianjun Zhang [4,5], Weiman Xing[1], Linchuan Liu[4,5] & Jianming Li [4,5,6]

Endoplasmic reticulum-associated degradation (ERAD) is a unique mechanism to degrade misfolded proteins via complexes containing several highly-conserved ER-anchored ubiquitin ligases such as HMG-CoA reductase degradation1 (Hrd1). Arabidopsis has a similar Hrd1-containing ERAD machinery; however, our knowledge of this complex is limited. Here we report two closely-related Arabidopsis proteins, Protein Associated With Hrd1-1 (PAWH1) and PAWH2, which share a conserved domain with yeast Altered Inheritance of Mitochondria24. PAWH1 and PAWH2 localize to the ER membrane and associate with Hrd1 via EMS-mutagenized Bri1 Suppressor7 (EBS7), a plant-specific component of the Hrd1 complex. Simultaneously elimination of two PAWHs constitutively activates the unfolded protein response and compromises stress tolerance. Importantly, the *pawh1 pawh2* double mutation reduces the protein abundance of EBS7 and Hrd1 and inhibits degradation of several ERAD substrates. Our study not only discovers additional plant-specific components of the Arabidopsis Hrd1 complex but also reveals a distinct mechanism for regulating the Hrd1 stability.

[1] Shanghai Center for Plant Stress Biology, Chinese Academy of Sciences, 201602 Shanghai, China. [2] University of Chinese Academy of Sciences, 100004 Beijing, China. [3] Shanghai Institute of Plant Physiology and Ecology, The Center of Excellence for Molecular Plant Sciences, Chinese Academy of Sciences, 300 Fenglin Road, 200032 Shanghai, China. [4] Guangdong Key Laboratory for Innovative Development and Utilization of Forest Plant Germplasm, College of Forestry and Landscape Architecture, South China Agricultural University, 510642 Guangzhou, China. [5] State Key Laboratory for Conservation and Utilization of Subtropical Agro-Bioresources, South China Agricultural University, 510642 Guangzhou, China. [6] Department of Molecular, Cellular, and Developmental Biology, University of Michigan, Ann Arbor, MI 48109-1048, USA. [7] These authors contributed equally: Liangguang Lin, Congcong Zhang. Correspondence and requests for materials should be addressed to L.L. (email: lcliu@scau.edu.cn) or to J.L. (email: jian@umich.edu)

Endoplasmic reticulum-associated degradation (ERAD) is an integral part of the ER-mediated protein quality control system, which constantly monitors the folding status of secretory and membrane proteins, repairs misfolding proteins, and degrades irreparable terminally misfolded proteins[1]. ERAD is a highly conserved degradation mechanism that involves substrate recognition, ubiquitination at the cytosolic surface of the ER membrane, retrotranslocation through ER membrane-embedded retrotranslocons, and eventual degradation by cytosolic proteasome[2]. The ERAD machinery builds around several ER membrane-anchored ubiquitin (E3) ligases that recognize different types of ERAD clients carrying structural defects in their luminal domains, transmembrane domains, or cytosolic domains (known as ERAD-L, ERAD-M, and ERAD-C substrates, respectively)[3]. One of the well-studied ERAD system is a multiprotein complex centered around an ER membrane-anchored ubiquitin E3 ligase known as HMG-CoA reductase degradation1 (Hrd1) in yeast[4] [HRD1 and glycoprotein 78 (gp78) in mammals[5,6]]. The Hrd1 complex is known to degrade ERAD-L and ERAD-M substrates and contains several other highly conserved membrane and luminal proteins[3]. They include Hrd3[7] [Sel1L (Suppressor of lin-12-like) in mammals[8]], Yos9 (Yeast OS-9 homolog)[9,10] [OS-9 (Osteosarcoma amplified 9), and XTP3-B (XTP3-transactivated protein B) in mammals[11,12]], Usa1[13] [U1 SNP1-associated protein 1; HERP for Homocysteine-induced ER Protein in mammals[14]], Der1 (Degradation in the endoplasmic reticulum1)[15] [DERLIN1-3 (Der-like domain-containing 1-3) in mammals[16]]. The Hrd1 complex also includes one or more ubiquitin-conjugating enzyme (E2), such as the ER membrane-anchored UBC6 (Ubiquitin-Conjugating 6) [Ube2j2 (Ubiquitin-conjugating enzyme E2 j2) in mammals[17]] and a cytosolic E2 UBC7 (Ube2g2 in mammals[18]) with its ER membrane-anchored recruiter Cue1 (Coupling of Ubiquitin conjugation to ER degradation that has no mammalian homolog)[19]. Biochemical and genetic studies in yeast and mammalian cells have shown that terminally misfolded glycoproteins with a unique asparagine-linked glycan (N-glycan) structure carrying an exposed $\alpha 1,6$-mannose residue are recruited to Hrd1 through a bipartite recruitment mechanism that uses Yos9/Os-9 to bind the $\alpha 1,6$-mannose-exposed N-glycans and Hrd3/Sel1L to bind exposed hydrophobic amino acid patches of misfolded glycoproteins[20]. A recruited ERAD client is subsequently ubiquitinated and retrotranslocated, which is likely mediated by Hrd1 in yeast[21], into the cytosol where the ubiquitinated ERAD substrate is degraded by the 26S proteasome. In both yeast and mammalian cells, deleting Hrd3/Sel1L significantly reduces the stability of Hrd1/HRD1[7,22,23]. Interestingly, while the $\Delta hrd3$-induced autodegradation requires Usa1 in yeast cells[24], the loss-of-Sel1L-caused HRD1 instability does not involve HERP in mammalian cells[23,25].

Recent studies have revealed that Arabidopsis has a similar Hrd1-mediated ERAD machinery[26]. This system is known to degrade two ER-retained mutant forms of the brassinosteroid (BR) receptor BRASSINOSTEROID-INSENSITIVE 1 (BRI1), bri1-5 and bri1-9[27,28], a misfolded conform of an Arabidopsis immunity receptor EF-Tu Receptor (EFR) produced in an Arabidopsis mutant defective in an ER-luminal protein folding sensor[29,30], several engineered ERAD substrates[31–34], and an ERAD component[35]. These studies not only identified conserved ERAD components in Arabidopsis, such as Hrd1a and Hrd1b[35,36], EBS5 (EMS-mutagenized bri1 suppressor 5; also known as HRD3A)[34,36], EBS6 (also known as AtOS9)[37,38], and UBC32 (Ubiquitin Conjugase 32)[33], which are homologs of the yeast Hrd1, Hrd3, Yos9, and Ubc7, respectively, but also discovered a plant-specific component known as EBS7 that regulates the protein stability of Hrd1a[39]. Analysis of the Arabidopsis genome fails to discover Arabidopsis homologs of Cue1 and Usa1

but identified three Arabidopsis homologs of Der1 [(Der1, Der2.1, and Der2.2)[40]. Compared to what has learnt from the yeast and mammalian systems, our knowledge of the plant Hrd1 ERAD complex remains limited[26]. For example, little is known about the composition and organization of the Arabidopsis Hrd1-complex, and it remains to be determined if the Arabidopsis Hrd1 ERAD complex contains a Der1/Derlin homolog. We also know little about how the protein stability and biochemical activity of Hrd1 are regulated in plants and whether or not the Arabidopsis Hrd1 is also involved in retrotranslocating its ERAD clients. In order to expand our understanding of the plant ERAD mechanism, we took a proteomic approach with independently generated transgenic lines expressing epitope-tagged Hrd1a/EBS7 to identify proteins that interact with both EBS7 and Hrd1a. Our subsequent biochemical and genetic studies demonstrated that two paralogous Arabidopsis proteins, named hereinafter as PAWH1 and 2 for Protein Associated With Hrd1, are crucial core components of the Arabidopsis Hrd1-containing ERAD machinery and are required to maintain the protein stability of both EBS7 and Hrd1a.

## Results

**Identification of PAWH1 and PAWH2 via a proteomic approach.** To identify additional components of the Arabidopsis Hrd1-complex, we employed a proteomic approach of immunoprecipitation coupled with mass spectrometry. We generated several transgenic lines expressing a fusion protein of green fluorescent protein (GFP) with the Arabidopsis Hrd1a (Hrd1a-GFP)[36], one of the two Arabidopsis Hrd1 homologs, or MYC/HA-tagged EBS7, a newly identified plant-specific component of the Arabidopsis ERAD machinery crucial for maintaining the Hrd1a stability[39]. Our previous studies have shown that simultaneous elimination of the two Arabidopsis Hrd1 homologs, Hrd1a and Hrd1b, or loss-of-function mutations in EBS7, inhibit ERAD of bri1-5 and bri1-9, two ER-retained mutant variant of the BR receptor BRI1 carrying a Cys[69]-Tyr mutation and a Ser[662]-Phe mutation in its extracellular domain, respectively[28,41]. Consequently, a small percentage of ER-accumulated bri1-5 and bri1-9 proteins leak out of the ER, likely due to saturation of their retention mechanisms, to reach the PM where the two mutant receptors bind extracellular BRs to promote plant growth, resulting in phenotypic suppression of the corresponding bri1-5 and bri1-9 dwarf mutants[36,39]. The three transgenes were able to rescue the corresponding hrd1a hrd1b bri1-9 and ebs7-3 bri1-5 mutants (Supplementary Fig. 1), respectively, indicating that all three tagged proteins are physiologically functional. We used one representative transgenic line for each transgene to extract total proteins or microsomal proteins and subsequently performed immunoprecipitation (IP) experiments with antibody-conjugated beads. The resulting immunoprecipitates were analyzed by liquid chromatography coupled with tandem mass spectrometry (LC-MS/MS) to identify proteins that were coimmunoprecipitated with the GFP-fused Hrd1a or MYC/HA-tagged EBS7. We also included the non-transgenic wild-type plant as our negative control to eliminate proteins that bound non-specifically to antibody-conjugated beads. Comparison of the five sets of coimmunoprecipitated proteins identified nine common proteins (Fig. 1a, b), including Hrd1a, EBS7, and a previously demonstrated component of the Arabidopsis ERAD machinery, EBS5[36] (also known as HRD3A[34] or SEL1L[38] that is the Arabidopsis homolog of the yeast Hrd3 and mammalian Sel1L). It is interesting to note that the three IP experiments with total proteins also identified Hrd1b and EBS6 (the Arabidopsis homolog of Yos9/OS-9[37,38]; Supplementary Fig. 2), suggesting the presence of multimeric Hrd1 in the Arabidopsis Hrd1 complex. The

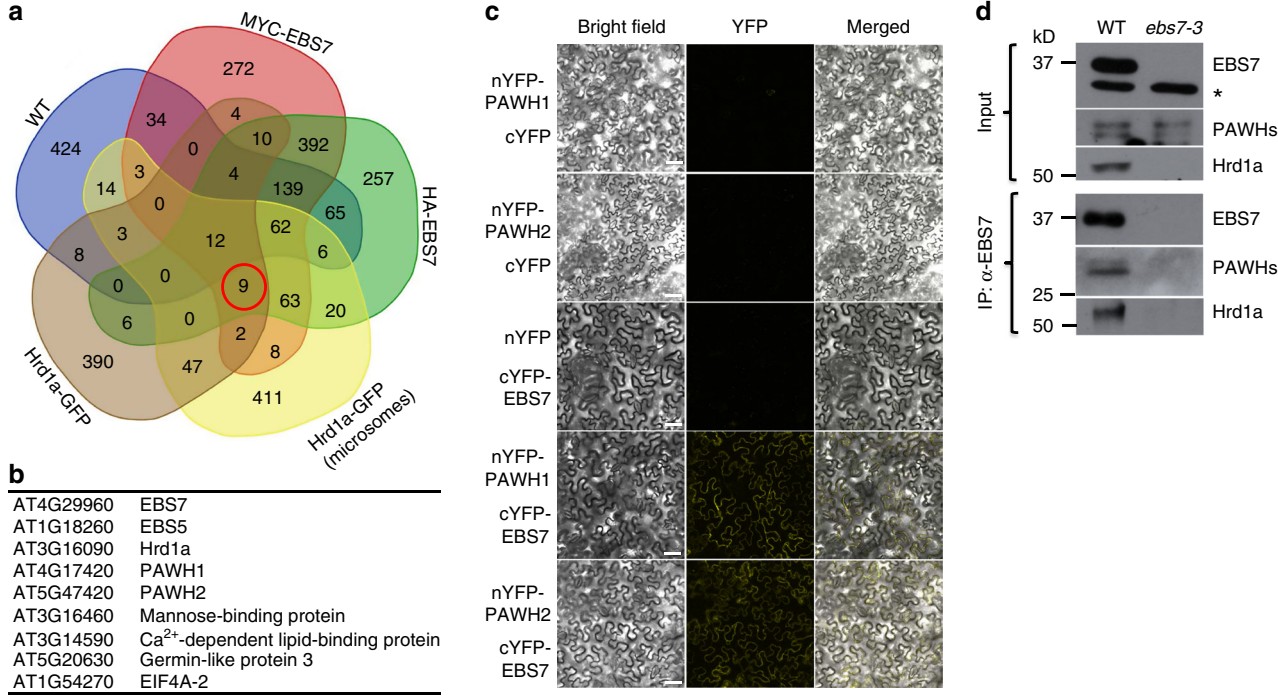

**Fig. 1** Discovery and verification of the two EBS7/Hrd1a-interacting PAWHs. **a** A Venn diagram of the immunoprecipitated proteins from total/microsomal proteins of the wild-type control or transgenic mutants expressing Hrd1a-GFP or MYC/HA-tagged EBS7. **b** The list of the nine proteins that were identified in all four IP-MS experiments. **c** The BiFC analysis of the PAWH1/2-EBS7 interaction in tobacco leaf epidermal cells. Bar = 50 μm. **d** CoIP of EBS7 with PAWH1/2 and Hrd1a in Arabidopsis plants. The total proteins (Input) and anti-EBS7 immunoprecipitates (IP:α-EBS7) were separated by SDS-PAGE and analyzed by immunoblotting with antibodies to EBS7, PAWH, and Hrd1a. The star indicates a non-specific cross-reactive band used to control equal amounts of proteins used for the coIP assays. The positions of molecular mass standards are indicated on the left. The raw data of the IP-MS experiments can be accessed at https://www.ebi.ac.uk/pride/archive with the dataset identifier PXD013400, and other source data are provided as a Source Data file

identification of known ERAD components as the abundant interacting proteins of both Hrd1a and EBS7 indicated success of our proteomic approach, whereas the failure to detect Hrd1b and EBS6 in the anti-GFP immunoprecipitates of the microsomal preparation was likely caused by low recovery of the immunoprecipitated proteins, evidenced by lower coverage of Hrd1a and EBS5 compared to a similar coimmunoprecipitation (coIP) experiment using the total proteins (Supplementary Fig. 2). Our analysis also identified two highly homologous proteins, At4g17420 (285 amino acids) and At5g47420 (282 amino acids) that were previously annotated as tryptophan RNA-binding attenuator protein-like proteins (TRAPs) and were renamed hereinafter as Protein Associated With Hrd1-1 (PAWH1) and PAWH2, respectively (Fig. 1b and Supplementary Fig. 2). A simple BLAST search revealed that PAWH1 and PAWH2 are highly conserved in the plant kingdom and contain AIM24 domain (Supplementary Fig. 3), which is originally discovered in the yeast mitochondria AIM24 (Altered Inheritance of Mitochondria protein 24) recently implicated in stabilizing the mitochondria contact site complex and the respiratory chain supercomplexes[42]. The four other proteins recovered in all 4 IP-MS experiments include a jacalin-related lectin (At3G16460), a calcium-dependent lipid-binding protein (At3g14590), a germin-like protein (At5g20630), and one of the three Arabidopsis translational initiation factor EIF4As (At1g54270). Further studies are needed to determine whether they are bona fide components of the Arabidopsis Hrd1 complex.

**Analysis of PAWH interactions with EBS7 and Hrd1a.** To test if PAWHs directly interact with Hrd1a and EBS7, we performed three experiments. The first one was a simple yeast two-hybrid assay with the predicted AIM24 domain-containing N-terminal 260 amino acids (AAs) of PAWH1 (PAWH1-N260) or 257 AAs of PAWH2 (PAWH2-N257) and the N-terminal soluble domain of EBS7 containing its N-terminal 142 AAs (EBS7-N142) or the cytoplasmic RING finger domain of Hrd1a (Hrd1a-CD). These assays showed that while PAWH1-N260 (also PAWH2-N257) interacted well with the EBS7-N142 fragment, it failed to interact with Hrd1a-CD that was previously shown to interact with the EBS7-N142 fragment[39] (Supplementary Figs. 4 and 5a). The second experiment was a transient bimolecular fluorescence complementation assay (better known as BiFC[43]) in tobacco leaf epidermal cells using the full-length PAWH1 and PAWH2 fused at their N-termini with the N-terminal half of the yellow fluorescent protein (nYFP-PAWH1 and nYFP-PAWH2) and the full-length EBS7 fused at its N-terminus with the C-terminal half of YFP (cYFP-EBS7) or the full-length Hrd1a fused with cYFP at its C-terminus (Hrd1a-cYFP). While yellow fluorescent signals were detected in tobacco leaf cells coexpressing nYFP-PAWH1 or nYFP-PAWH2 with cYFP-EBS7 (Fig. 1c), no fluorescent signal was detected in tobacco leaf cells coexpressing nYFP-PAWH1 or nYFP-PAWH2 with Hrd1a-cYFP (Supplementary Fig. 5b). In addition, we performed a coIP experiment with total proteins from the wild-type Arabidopsis seedlings using anti-EBS7 antibody[39], a custom-made anti-Hrd1a antibody (Supplementary Fig. 6), and a custom-made anti-PAWH antibody that could identify both PAWH1 and PAWH2 proteins (see Supplementary Fig. 11d). As shown in Fig. 1d, the anti-EBS7 antibody not only immunoprecipitated the endogenous EBS7 protein but also brought down Hrd1a and the two PAWH proteins. By contrast, neither Hrd1a nor PAWHs were detected in a similar coIP assay with the total proteins extracted from the null *ebs7-3* mutant[39] (Fig. 1d). These results suggested that PAWH1/2 could directly

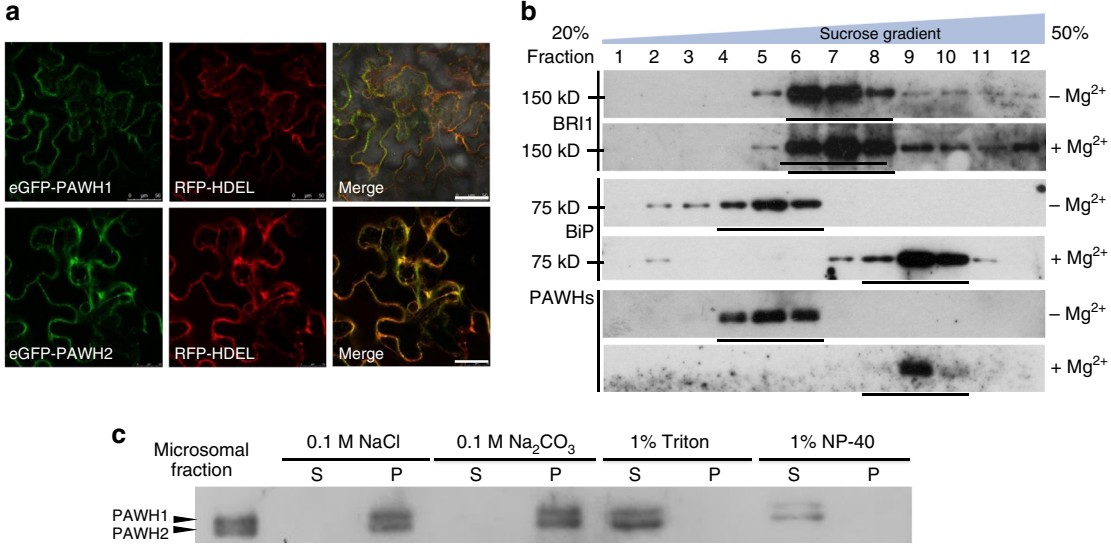

**Fig. 2** PAWHs are ER membrane-anchored proteins. **a** The confocal microscopic images of tobacco leaf epidermal cells that transiently express GFP-PAWH1/2 (left), the ER-localized RFP-HDEL (middle), and merged images of green and red fluorescent signals (right). Bar = 50 μm. **b** Sucrose gradient ultracentrifugation of PAWH1/2 proteins/protein complexes. Protein samples from collected gradient fractions of a linear (20–50%) sucrose gradient in the presence (+) or absence (−) of $Mg^{2+}$ were separated by SDS-PAGE and analyzed by immunoblotting with antibodies to BRI1, BiPs, and PAWH. The positions of molecular mass standards are shown on the left. **c** Solubilization of PAWHs by various solvents/detergents. The microsomal preparations of 10-day-old Arabidopsis seedlings were treated with 0.1 M NaCl, 0.1 M $Na_2CO_3$, 1% (v/v) Triton X-100, or 1% (v/v) Nonidet P-40 for 4 h and centrifuged at 100,000×g for 60 min to generate both supernatant (S) and pellet (P) fractions, which were subsequently separated by SDS-PAGE and analyzed by immunoblotting with anti-PAWH antibody. Source data are provided as a Source Data file

bind EBS7 but not Hrd1a and that the coIP-detected Hrd1a-PAWH1/2 association is likely mediated by EBS7 known to interact directly with Hrd1a[39].

**PAWHs are ER membrane proteins and induced by ER stress.**
Both *PAWH* genes are widely expressed in various tissues/organs throughout the Arabidopsis development with *PAWH1* exhibiting high expression at later stages of seed development and reaching its highest expression level in dry seed (Supplementary Fig. 7). The two *PAWH* genes were previously shown to coexpress with genes known/predicted to be involved in protein folding and/or protein quality control (Supplementary Fig. 8)[44]. Many *PAWH*-coexpressed genes were known to be induced by ER stress[45]. To test if *PAWH* transcripts and PAWH proteins are also induced by ER stress, we treated the wild-type seedlings with or without tunicamycin (TM), a widely used ER stress inducer that inhibits the first biosynthetic step of the N-glycan precursor, and used the treated seedlings to examine the abundance of *PAWH* transcripts and PAWH proteins by RT-PCR and immunoblotting, respectively. We found that the TM treatment increased the mRNA levels of both *PAWH* genes and elevated the protein abundance of both PAWHs (Supplementary Fig. 9).

Both PAWHs lack the N-terminal signal peptide or the C-terminal ER retrieval motifs but were predicted to contain a potential transmembrane (TM) segment near their C-termini (Supplementary Fig. 10). To directly determine their subcellular localization, we generated GFP-fusion transgenes for the two *PAWH* genes and transiently expressed them in tobacco leaf epidermal cells. Confocal microscopy analysis of agro-infiltrated tobacco leaves revealed that the green fluorescent signals of the GFP-PAWH1/2 fusion proteins overlapped with the fluorescent signals of the red fluorescent protein (RFP) tagged at its C-terminus with the HDEL (histidine-aspartate-glutamate-leucine) ER retrieval motif (Fig. 2a), a widely used ER-localized marker, strongly suggesting that PAWH1/2 are localized to the ER. A further confirmation was provided by sucrose density-gradient

centrifugation with Arabidopsis microsomal proteins in the presence or absence of $Mg^{2+}$. Because $Mg^{2+}$ is required for polyribosome binding to the ER and that the density of the ribosome-bound ER membrane is higher than that of the ribosome-free ER or other microsomal membranes, an ER-localized protein should undergo a diagnostic $Mg^{2+}$-dependent shift from lower density to higher density on a sucrose gradient[46]. Figure 2b shows that PAWHs exhibited a similar $Mg^{2+}$-dependent density shift as BiPs (binding immunoglobulin proteins, ER-localized Heat Shock Protein 70), but differs from BRI1 known to be localized to the PM[47]. Consistent with the predicted C-terminal TM segments (Supplementary Fig. 10), immunoblot analysis of PAWHs in soluble and insoluble fractions of resuspended Arabidopsis microsomal pellets in 0.1 M NaCl, 0.1 M $Na_2CO_3$, 1% (v/v) Triton X-100, or 1% (v/v) Nonidet P-40 solution showed that only the two nonionic detergents could release the two PAWHs from microsomes, suggesting that both PAWHs are likely anchored to the ER membrane (Fig. 2c).

**Mutating both PAWHs stabilizes several ERAD substrates.** To investigate if the two PAWHs are involved in an Arabidopsis ERAD process, we obtained T-DNA insertional mutants for the two *PAWH* genes from the Arabidopsis Biological Resource Center (http://abrc.osu.edu/)[48] (Supplementary Fig. 11a, b). RT-PCR analysis indicated that the two mutants failed to produce the full-length transcript of the corresponding *PAWH* gene while immunoblot analysis failed to detect one of the two crossing-reacting bands of the wild-type plant (Supplementary Fig. 11c, d), indicating that the two T-DNA insertional mutants are null mutants. We crossed each mutant into *bri1-9* and found that the resulting *pawh1 bri1-9* and *pawh2 bri1-9* mutants were morphologically indistinguishable from *bri1-9* (Fig. 3a–c). A further cross between the two double mutants generated a *pawh1 pawh2 bri1-9* triple mutant that was morphologically similar to the wild-type (Fig. 3a–c), indicating that simultaneous elimination of PAWH1 and PAWH2 suppressed the *bri1-9* dwarf phenotype.

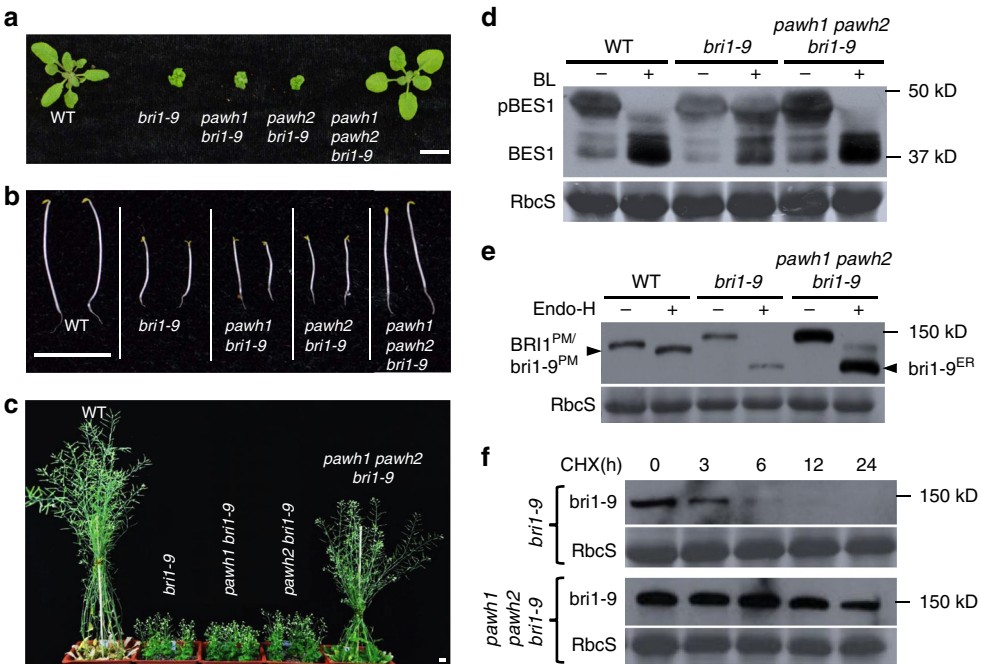

**Fig. 3** Mutating two PAWHs suppresses the *bri1-9* phenotype and inhibits bri1-9 degradation. **a–c** Photographs of 3-week-old light-grown seedlings (**a**), 7-day-old dark-grown seedlings (**b**), and 2-month-old mature soil-grown plants (**c**). Scale bar = 1 cm. **d–f** Immunoblot analysis of BR-triggered BES1 dephosphorylation (**d**), the Endo H sensitivity of BRI1/bri1-9 (**e**), and the protein stability of bri1-9 (**f**). Total proteins were extracted from 10-day-old Arabidopsis seedlings treated with (+) or without (−) 1 μM brassinolide (BL, the most active member of the BR family) for 1 h (**d**) or 180 μM CHX (**f**) for indicated durations. Proteins extracted from non-treated seedlings were treated with (+) or without (−) Endo H (**e**). These protein samples were separated by SDS-PAGE and analyzed by immunoblotting with antibodies to BES1 (**d**) or BRI1 (**e**, **f**). The lower strips in **d–f** showing the Ponceau Red-stained small subunit of the ribulose-1,5-bisphospate carboxylase/oxygenase (RbcS), were used as loading controls. The positions of the molecular mass standards are shown on the right. In **e** BRI1[PM]/bri1-9[PM] and bri1-9[ER] denote BRI1/bri1-9 proteins localized on the PM and in the ER, respectively. Source data are provided as a Source Data file

Consistent with the phenotypic suppression, the *pawh1 pawh2* double mutation partially restored the BR sensitivity to the *bri1-9* mutant revealed by the BR-triggered root growth inhibition assay[49] (Supplementary Fig. 12) and the BR-induced BES1 dephosphorylation assay[50] (Fig. 3d). Importantly, immunoblot analysis showed that the *pawh1 pawh2* double mutation greatly increased the bri1-9 abundance, which was much higher than the abundance of the wild-type BRI1 that undergoes a phosphorylation-dependent endocytosis and ubiquitin-mediated degradation[51]. We also treated total proteins with endoglycosidase H (Endo H) that cleaves high mannose-type N-glycans of ER-localized/retained glycoproteins but does not cut Golgi-processed N-glycans of glycoproteins. We found that the *pawh1 pawh2* mutation increased abundance of the Endo H-resistant form of bri1-9 (Fig. 3e), suggesting increased presence of bri1-9 on the PM to promote growth. Indeed, our transgenic experiment showed that overexpression of EBS2 known to retain bri1-9 in the ER[52], significantly decreased the amount of the Endo H-resistant form of bri1-9 and nullified the suppressive effect of the *pawh1 pawh2* double mutation on the *bri1-9* dwarfism while having a little effect on the total amount of bri1-9 proteins (Supplementary Fig. 13). To determine the biochemical basis of increased bri1-9 abundance in the *pawh1 pawh2 bri1-9* mutant, we performed a cycloheximide (CHX)-chasing experiment and found that increased bri1-9 abundance is caused by decreased protein degradation rather by increased protein biosynthesis (Fig. 3f). A subsequent rescue experiment showed that the effect on growth and biochemical phenotypes were indeed caused by the double *pawh1 pawh2* mutation (Supplementary Fig. 14). Taken together, these experiments demonstrated that PAWH1 and PAWH2 are

crucial components of the Arabidopsis ERAD machinery that degrades the ER-retained mutant bri1-9 receptor.

A similar set of experiments revealed that simultaneous elimination of the two PAWHs by the T-DNA insertional mutations or by CRISPR/Cas9-mediated genome editing inhibited the degradation of bri1-5 and suppressed the dwarf phenotype of the corresponding *bri1-5* mutant (Supplementary Figs. 15 and 16). We also tested if the *pawh1 pawh2* double mutation could inhibit degradation of other misfolded Arabidopsis proteins. Previous studies showed that the Arabidopsis EF-Tu receptor (EFR), a cell surface-localized receptor that detects and binds a small peptide derived from the highly conserved translation factor EF-TU (elongation factor thermo unstable) of pathogenic bacteria[53], becomes misfolded and degraded in an Arabidopsis mutant defective in an ER-luminal protein homologous to the mammalian UDP-glucose:glycoprotein glucosyltransferase[29,30] (also known as EBS1[41]). We crossed the *pawh1 pawh2* mutation into an *ebs1* mutant and analyzed the EFR abundance in *ebs1*, *pawh1 pawh2*, *ebs1 pawh1 pawh2*, and their wild-type control. Consistent with the previous findings[29,30], EFR was non-detectable in *ebs1* but its level was greatly increased in the *ebs1 pawh1 pawh2* triple mutant (Supplementary Fig. 17), demonstrating that the two PAWHs are also involved in degrading a misfolded EFR. We thus concluded that PAWH1 and PAWH2 are likely general components of the Arabidopsis Hrd1-containing ERAD complex.

**The loss of PAWHs activates UPR and reduces stress tolerance.** Previous studies have shown that loss-of-function ERAD

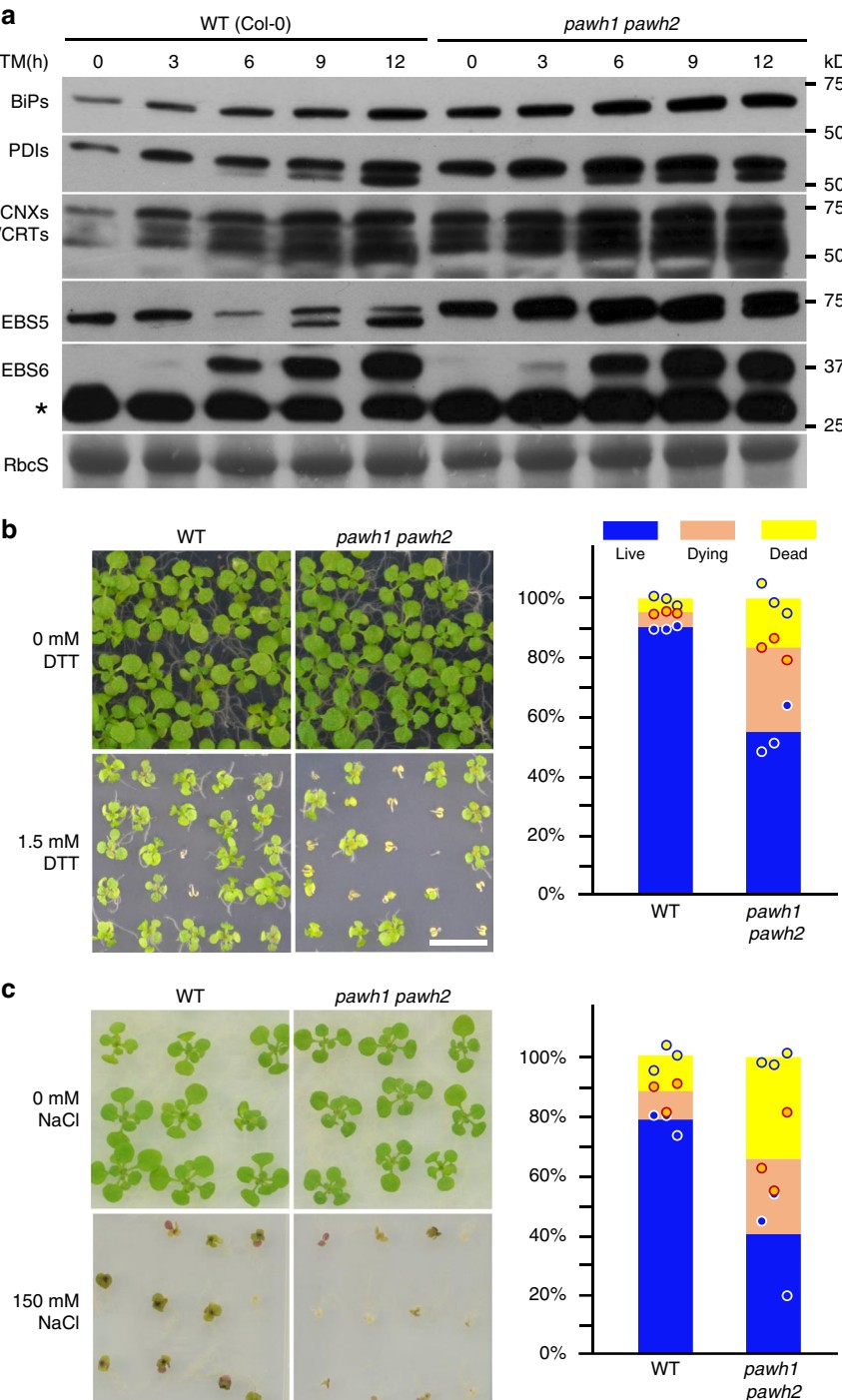

**Fig. 4** The *pawh1 pawh2* mutation activates UPR and reduces stress tolerance. **a** Immunoblot analysis of various ER-localized proteins in seedlings treated with TM for different durations. The faster moving bands detected by anti-PDI and anti-EBS5 antibody are non-glycosylated forms of PDIs and EBS5, respectively. The star indicates a non-specific band. The lower strip is the Ponceau Red-stained RbcS bands used as the loading control. **b**, **c** The left are photographs of 15-day-old Arabidopsis seedlings grown on ½ MS medium supplemented with or without 1.5 mM DTT (**b**) or 150 mM NaCl (**c**), while the right are the bar graphs showing the percentage of three kinds of seedlings: blue (alive), brown (dying), and yellow (dead). Scale bar = 1 cm. The stress sensitivity experiments were repeated three times with ~100 seedlings/each for the DTT assay and ~50 seedlings/each for the NaCl experiment with individual data points shown as open circles distributed above or below the average values of the three repeats. Source data are provided as a Source Data file

mutations often result in constitutive activation of the Arabidopsis unfolded protein response (UPR) pathway[36–39], a highly conserved ER stress response pathway that upregulates production of ER chaperones and ERAD components to maintain proteostasis[54,55]. To examine if the *pawh1 pawh2* mutation also activates UPR, we performed an immunoblot analysis with total proteins of the wild-type and *pawh1 pawh2* mutant treated with or without TM for variable durations. Figure 4a shows that the protein abundance of BiPs, protein disulfide isomerases (PDIs), calreticulins/calnexins (CRT/CNXs), EBS5, and EBS6 was higher

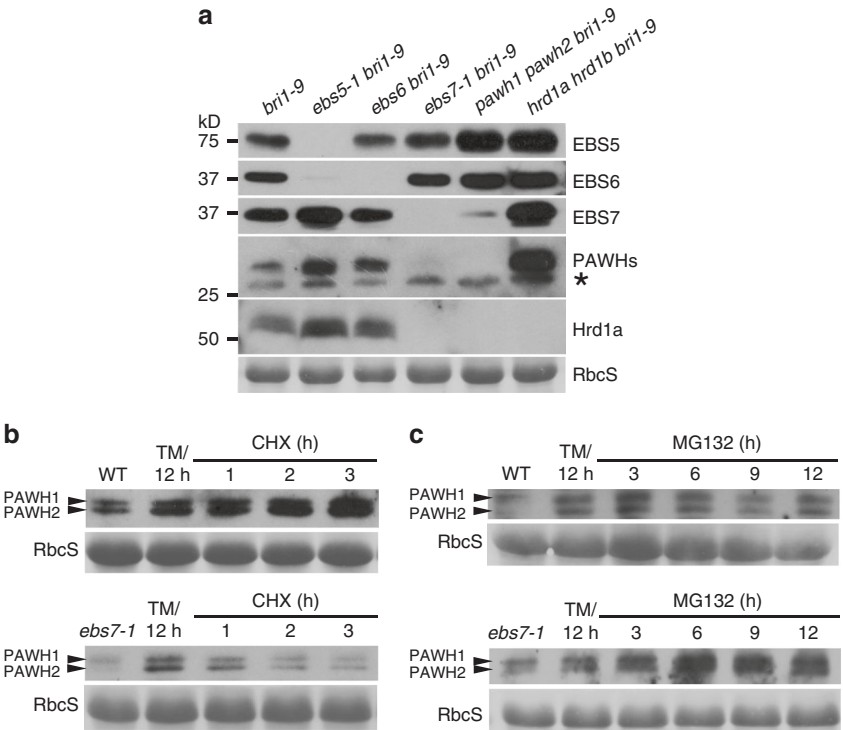

**Fig. 5** Impact of ERAD mutations on the stability of EBS7, PAWHs, and Hrd1a. **a** Immunoblot analysis of the protein abundance of EBS5, EBS6, EBS7, PAWH1/2, and Hrd1a in 4-week-old seedlings of various ERAD mutants. Asterisk indicates a non-specific band. **b**, **c** Immunoblot analysis of the protein abundance of PAWHs. Total proteins extracted from 10-day-old seedlings treated with 5 μg/mL TM for 12 h followed by treatment with 180 μM CHX (**b**) or 80 μM MG132 (**c**) for different durations were separated by SDS-PAGE and analyzed by immunoblotting with anti-PAWH antibody. In **a–c**, the RbcS-labeled strips show the Ponceau Red-stained RbcS band and were used as the loading controls for the experiments. Source data are provided as a Source data file

in *pawh1 pawh2* seedlings treated with or without TM than the corresponding wild-type seedlings, indicating that the *pawh1 pawh2* mutation constitutively activates the UPR pathway. The detected enhancement of UPR prompted us to test if the *pawh1 pawh2* double mutation affects the plant stress tolerance, which is known to involve the UPR pathway[55]. We germinated seeds and grew seedlings on ½ MS medium containing 1.5 mM dithiolthreitol (DTT, a widely used ER stress-inducer that interferes with oxidative protein folding) and 150 mM NaCl known to cause salt stress in Arabidopsis. As shown in Fig. 4b, c, the percentages of dying and dead seedlings on DTT/NaCl-containing medium were markedly higher for the *pawh1 pawh2* mutant than the wild-type control, indicating that the *pawh1 pawh2* double mutation compromises the plant stress tolerance.

**Both PAWH1 and PAWH2 are unstable in the *ebs7-1* mutant.** Given the facts that EBS7 interacts with both Hrd1a and PAWHs and that *ebs7* mutations greatly destabilize Hrd1a[39], we were interested in examining the impact of the *ebs7-1* mutation on the PAWH1/2 abundance as well as the impact of the *pawh1 pawh2* mutation on the protein stability of both EBS7 and Hrd1a. We also included three other ERAD mutants, *ebs5 bri1-9*, *ebs6 bri1-9*, and *hrd1a hrd1b bri1-9* in our experiment. Consistent with our earlier finding[39], the missense *ebs7-1* mutation (Ala[131]-Thr) caused disappearance of not only the mutated ebs7-1 protein but also Hrd1a (Fig. 5a). Interestingly, *ebs7-1* also markedly reduced the protein abundance of the two PAWHs (Fig. 5a). By contrast, the *ebs6* mutation seemed to have a marginal effect on the abundance of EBS7, PAWHs, or Hrd1a, whereas the *ebs5* mutation increased the protein levels of EBS7 and PAWHs (Fig. 5a), which could be caused by non-significant and significant impact

of the *ebs6* and *ebs5* mutations, respectively, on the transcript abundance of *EBS7* and *PAWHs* (Supplementary Fig. 18). To determine if the *ebs7-1*-caused reduction of PAWH abundance is attributed to increased protein degradation, we first treated seedlings of wild-type and *ebs7-1* with TM for 12 h to increase the PAWH1/2 abundance, subsequently treated these seedlings with CHX or MG132 (a widely used proteasome inhibitor) for different durations, and analyzed the PAWH abundance by immunoblotting. Figure 5b shows that while the CHX treatment had little impact on the PAWH1/2 abundance in wild-type seedlings, the same treatment led to a rapid decrease in the PAWH1/2 abundance in *ebs7-1* seedlings. Consistently, a 12-h-treatment of MG132 caused little change or slightly reduction in the PAWH abundance in wild-type seedlings, whereas the same MG132 treatment actually increased the PAWH1/PAWH2 protein levels in the *ebs7-1* mutant (Fig. 5c). Together, these experiments indicated that *ebs7-1* destabilizes the two PAWH proteins that are likely degraded via a proteasome-mediated process.

**The *pawh1 pawh2* mutation destabilizes EBS7 and Hrd1a.** Our immunoblot assays of various ERAD mutants also revealed that the *pawh1 pawh2* double mutation greatly reduced the protein abundance of both EBS7 and Hrd1a (Fig. 5a), while our quantitative real-time RT-PCR (qPCR) analyses showed that the *pawh1 pawh2* double mutation slightly elevated the levels of the *EBS7* mRNA but had a marginal impact on the *Hrd1a* transcript abundance (Supplementary Fig. 18). To determine if the observed reduction in EBS7 and Hrd1a protein abundance is due to increased protein degradation or decreased protein synthesis, we used the same TM-CHX/MG132 protocol described above to

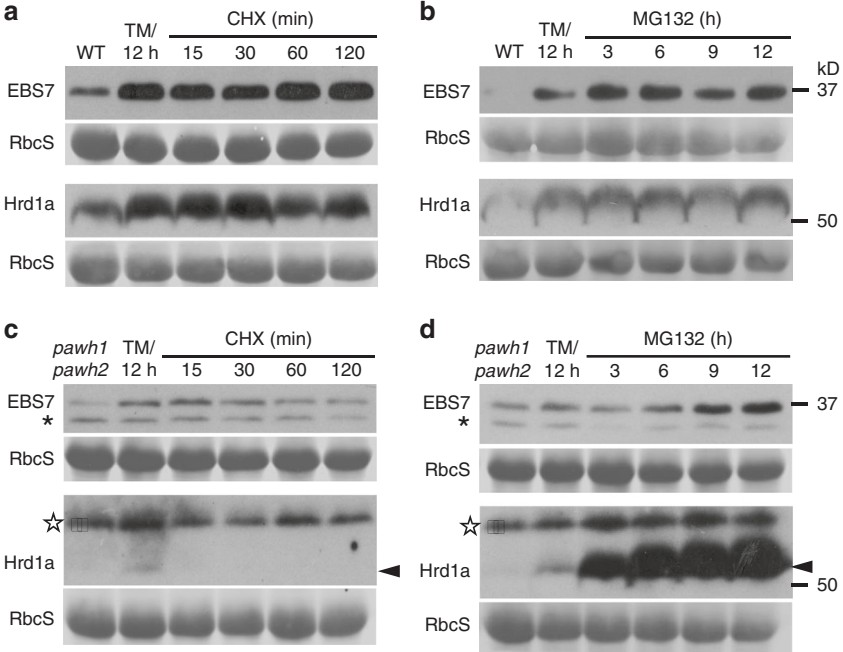

**Fig. 6** The *pawh1 pawh2* double mutation reduces the stability of EBS7 and Hrd1a. **a–d** Immunoblot analysis of EBS7 and Hrd1a in the wild-type (**a**, **b**) and *pawh1 pawh2* double mutant (**c**, **d**). Total proteins were extracted from 10-day-old seedlings that were initially treated with 5 µg/mL TM for 12 h and subsequently treated with 180 µM CHX (**a**, **c**) or 80 µM MG132 (**b**, **d**) for different durations, separated by SDS-PAGE, and analyzed by immunoblotting using anti-EBS7 and anti-Hrd1a antibodies. In each immunoblotting experiment, the RbcS-labeled strip was used as a loading control. Asterisks and stars denote cross-reacting bands to anti-EBS7 and anti-Hrd1a antibodies, respectively, while arrows indicate the Hrd1a band. Source data are provided as a Source Data file

treat seedlings of the wild-type and the *pawh1 pawh2* double mutant and analyzed the protein abundance of EBS7 and Hrd1a by immunoblotting. As shown in Fig. 6a, b, neither CHX or MG132 treatment had an observable effect on the protein abundance of EBS7 or Hrd1a in the wild-type seedlings. However, the CHX treatment caused a gradual reduction of EBS7 but very rapid disappearance of the TM-induced Hrd1a (Fig. 6c), whereas the MG132 treatment gradually stabilized EBS7 and caused rapid and significant stabilization of the ER-anchored E3 ligase (Fig. 6d). We concluded that the *pawh1 pawh2* double mutation destabilizes both EBS7 and Hrd1a that are likely degraded via a proteasome-mediated mechanism.

Because PAWHs fail to interact directly with Hrd1a (Supplementary Fig. 5) and the *pawh1 pawh2* mutation greatly reduces the protein abundance of EBS7 (Fig. 5a), which is known to be important to maintain the Hrd1a stability[39], we wondered whether the impact of the *pawh1 pawh2* mutation on Hrd1a stability is a secondary effect caused by reduced EBS7 abundance. We therefore overexpressed EBS7 using the strong *p35S* promoter to drive the expression of HA-tagged EBS7 in the *pawh1 pawh2 bri1-9* triple mutant. This experiment revealed that while the *p35S::HA-EBS7* transgene did lead to increased EBS7 protein abundance in the triple mutant, it failed to stabilize Hrd1a or decrease the bri1-9 abundance (Supplementary Fig. 19), suggesting that PAWHs exert their direct impact on the ER-anchored E3 ligase via a yet unknown mechanism.

Our immunoblot analysis also revealed that the *hrd1a hrd1b* double mutation did not reduce but increased the abundance of EBS5, EBS6, EBS7, and the two PAWH proteins (Fig. 5a). Because our qPCR experiments showed that the *hrd1a hrd1b* double mutation significantly increased the transcript levels of both *PAWH* genes but had a marginal impact on the *EBS7* transcript abundance (Supplementary Fig. 18), we suspected that Hrd1a and/or Hrd1b might be the E3 ligase that ubiquitinates EBS7 in

the *pawh1 pawh2* mutant to promote EBS7 degradation. Consistent with this hypothesis, overexpression of a GFP-tagged Hrd1a driven by the strong *p35S* promoter not only reduced the bri1-9 abundance and complemented the phenotypes of the *hrd1a hrd1b bri1-9* triple mutant, but also greatly lowered the protein levels of bri1-9, EBS7, and the two PAWHs (Supplementary Fig. 20). Because both EBS7 and PAWHs remained stable in the *hrd1a hrd1b* mutant background, we also used total proteins of the *hrd1a hrd1b bri1-9* mutant to analyze the EBS7-PAWHs interaction and discovered that the *hrd1a hrd1b* mutation had no detectable impact on the EBS7-PAWHs interaction (Supplementary Fig. 21).

## Discussion

In this study, we used a proteomic approach to identify two redundant proteins, PAWH1 and PAWH2, which were coimmunoprecipitated with GFP-tagged AtHrd1a and MYC/HA-tagged EBS7 expressed in their corresponding loss-of-function Arabidopsis mutants. Previous transcriptome analyses indicated that both *PAWH* genes were co-expressed with genes known or predicted to encode ER chaperone, folding catalysts, and ERAD components, while our current study confirmed that both PAWH1 and PAWH2 proteins were induced by chemically triggered ER stress. Despite the lack of an N-terminal signal peptide and known ER retention/retrieval sequence motifs, both PAWHs were found to be localized at the ER membrane, likely through its predicted C-terminal transmembrane segment, as evidenced by confocal microscopy, sucrose-gradient ultracentrifugation, and solubilization of microsomal proteins with different solvents and detergents.

In addition, our yeast two-hybrid assay, transient BiFC assay in tobacco leaves, and coIP experiments using transgenic Arabidopsis plants showed that PAWHs interact directly with EBS7 but

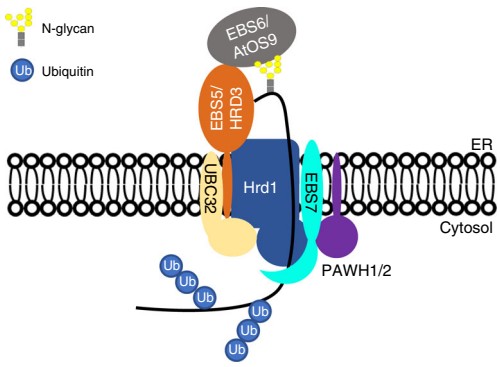

**Fig. 7** A model of the Arabidopsis Hrd1 complex. An ERAD-L substrate is recognized and recruited by both EBS6/AtOS9 and EBS5/HRD3 to the ER membrane-anchored Hrd1 that works with the E2 UBC32 to ubiquitinate the ERAD client. EBS7 and PAWH1/2 interact with Hrd1 to regulate the stability and activity of the E3 ligase

associate with Hrd1a indirectly, most likely mediated by the previously demonstrated EBS7-Hrd1a interaction[39]. Furthermore, we showed that the *pawh1 pawh2* double mutation constitutively activated the UPR pathway and greatly compromised plant tolerance against salt stress and DTT-triggered ER stress, similar to what were previously reported for Arabidopsis ERAD mutants[36–39]. More importantly, we have shown that the *pawh1 pawh2* mutation blocked degradation of two ER-retained mutant BR receptors, bri1-5 and bri1-9, allowing them to accumulate and consequently leak out of the ER to reach the PM where the two mutant BR receptors could respond to BRs to stimulate plant growth. Our study also showed that the *pawh1 pawh2* mutation also inhibited degradation of misfolded EFR in the *ebs1* mutant background, strongly suggesting that the two redundant PAWH proteins are likely general components of the Arabidopsis Hrd1-containing ERAD complex that degrades many misfolded proteins. Thus, our study, coupled with previously reported results, revealed that such an Arabidopsis ERAD complex consists of not only evolutionarily conserved components, including EBS5/HRD3A[34,36], EBS6/AtOS9[37,38], Hrd1a/1b[36], and their associated E2 conjugase UBC32[33], but also plant-specific components, such as EBS7[39] and PAWH1/PAWH2 (Fig. 7).

Our results have shown that the plant-specific components, EBS7 and PAWHs, are important to maintain the protein stability of Hrd1a. In the absence of the two homologous PAWH proteins, the stability of EBS7 and Hrd1 was significantly reduced. Our previous study demonstrated that *ebs7* mutations significantly reduce the Hrd1 stability[39], raising a question if reduction of Hrd1 stability by the *pawh1 pawh2* mutation is an indirect consequence of reduced abundance of EBS7 that directly binds PAWHs. However, our transgenic experiment showed that increased production of EBS7 driven by the *p35S* promoter failed to increase Hrd1a abundance, suggesting a direct role of PAWHs in maintaining the Hrd1a stability. This finding and our earlier results of the EBS7-AtHrd1a interaction demonstrated that EBS7 and PAWHs work together to regulate the stability of the ER membrane-anchored Hrd1a.

Studies in yeast and mammalian cells showed that Hrd3/Sel1L is essential to maintain the stability of Hrd1 and that mutations in Hrd3 in yeast or Sel1L in mammalian cells resulted in a significant reduction in the stability of the yeast or mammalian Hrd1 E3 ligase[7,22,23]. However, mutations of EBS5, the Arabidopsis homolog of Hrd3/Sel1L[36], had no detectable impact on the Hrd1a stability. Our results thus revealed a novel mechanism that regulates the protein stability/activity of a plant ERAD E3 ligase, which is quite diverged from a conserved regulatory mechanism

that controls the stability of yeast and mammalian Hrd1s. In yeast, the *hrd3Δ*-triggered Hrd1 loss is caused by autodegradation facilitated by the Hrd1's own catalytic RING domain[24]. It remains to be tested whether or not the disappearance of Hrd1a in *ebs7* or *pawh1 pawh2* mutant is caused by self-destruction of Hrd1a or by other ER-anchored E3 ligases such as SUPPRESSOR OF DRY2 DEFECTS1[56], an Arabidopsis homolog of the yeast Degradation of alpha factor2-10 (Doa10) that is known to be involved in degrading misfolded transmembrane proteins carrying a cytosolic structural defect[3] or one of the three Arabidopsis RING finger proteins with membrane anchor (RMA1-3)[57].

While loss-of-function mutations in EBS7 and PAWHs destabilize PAWHs/Hrd1a and EBS7/Hrd1a, respectively, the *hrd1a hrd1b* double mutation does not reduce but increases the protein abundance of both EBS7 and PAWHs. We hypothesize that this is caused by increased transcription and reduced protein degradation. The *hrd1a hrd1b* double mutation may cause accumulation of misfolded proteins in the ER and consequently activates the UPR pathway, which increases production of many chaperones, folding enzymes, and components of the ERAD machinery such as EBS7 and PAWHs. Moreover, the two functionally redundant Hrd1 homologs likely trans-ubiquitinate EBS7 to regulate its protein abundance via the so-called ER-tuning mechanism[35,58] to tightly and rapidly regulate the ERAD capacity to adapt to fluctuations in the amount of misfolded proteins accumulated in the ER, thus maintaining the ER proteostasis essential for cell survival. A further investigation is needed to fully understand the role of Hrd1 in such an ER-tuning regulatory mechanism.

Although our study demonstrated an essential role of PAWHs in maintaining the stability of Hrd1a, it remains unknown how the two PAWHs execute this function. Both PAWH1 and PAWH2 carry a highly conserved domain that was first discovered in the yeast AIM24 protein, a mitochondria protein tightly associated with the inner membrane and required for the ultrastructure, composition, and function of mitochondria[42,59]. It was previously thought that AIM24 is a fungal invention[60], but the 32nd release of the Pfam database of protein families in Sept. 2018 (https://pfam.xfam.org) revealed a total of 5937 AIM24 domain-containing proteins from 3272 species in all three domains of life, with >93% of them containing only the AIM24 domain (Supplementary Fig. 22). Despite wide occurrence of the AIM24 domain-containing proteins (a total of 4535 sequences) in bacteria, nothing is known about physiological function of any of these bacterial proteins. Interestingly, crystal structures of at least two bacterial AIM24 domain-containing proteins were solved: an unknown protein of 248 AAs (PA3696) from *Pseudomonas aeruginosa* strain PA01 (PDB: 1YOX), and a hypothetic protein of 243 AAs (SpyM3_0169) from *Streptococcus pyogenes* (PDB: 1PG6). Structural modeling at SWISS-MODEL server (https://swissmodel.expasy.org) using the structures of these two bacterial proteins as templates suggested that PAWH1/2 likely adopt a similar monomeric structure consisting of three structural repeats of beta-sandwich (Supplementary Fig. 23a, b), each having two beta-sheets of 3/4 anti-parallel beta-strands. It is worth mentioning that each structure unit resembles the crystal structure of the bacterial tryptophan RNA-binding attenuation protein (PDB: 1WAP)[61], explaining the original names of the two PAWHs as tryptophan RNA-binding attenuator-like proteins. Our structural modeling also suggested that PAWHs could form a homotrimeric subcomplex with a flat surface on the bottom (Supplementary Fig. 23c–f). It is important to note that the predicted C-terminal transmembrane α-helix is not included in the models and that the C-terminal α-helices shown in the two models correspond to the peptide fragment of 239-250 of PAWH1 (236-247 in PAWH2), which is unlikely a transmembrane domain due to its short length.

Such a modeled PAWH structure suggests that PAWHs likely rely on their flat ring-surface to interact with other proteins and use the protruding α-helices (corresponding to the consensus C-terminal transmembrane α-helical domain, see Supplementary Fig. 10) to attach the trimeric PAWH complex into the ER membrane. Further studies are needed to determine if the predicted C-terminal transmembrane domain is needed for the biological function of PAWHs, and more importantly, to identify additional PAWH-interacting proteins and/or crucial amino acid residues of PAWH1/2, which could help to understand the biochemical mechanism(s) by which PAWHs regulate the protein stability and catalytic activity of the plant Hrd1.

## Methods

**Plant materials and growth conditions.** Most of the Arabidopsis wild-type, mutants, and transgenic lines are in the Columbia (Col-0) ecotype except for mutants and transgenic lines that carry the *bri1-5* mutation, which are in the Wassilewskija-2 (Ws-2) ecotype. The Arabidopsis mutants used in this study include *bri1-5*[62], *bri1-9*[41], *ebs1*[41], *ebs5*[36], *ebs6*[37], *ebs7-1*[39], and *hrd1a hrd1b*[36]. The T-DNA insertional mutants CS335767 (*pawh1*) and SALK_111654 (*pawh2*) were obtained from Arabidopsis Biological Resource Center (ABRC) at Ohio State University[48,63]. The *pawh1-c* and *pawh2-c* mutants were created by CRISPR/Cas9-mediated genome editing in the *bri1-5* mutant background (see below for the experimental details). Seed were surface sterilized using the ethanol-washing protocol[64] and seedling were grown at 22 °C in growth chamber or growth room under long-day (16 h-light/8 h-dark) photoperiodic condition.

**Chemical treatment of Arabidopsis seedlings.** To study the stress tolerance, seeds were germinated, and young seedlings were grown on ½ Murashige and Skoog (MS, Sigma) medium supplemented with or without 1.5 mM DTT (Roche) or 150 mM NaCl for 15 days and the numbers of seedlings that were green (alive), yellowish (dying), or dead were counted and recorded. To analyze UPR activation, 10-day-old Arabidopsis seedlings were carefully transferred into liquid ½ MS medium supplemented with 5 μg/mL TM (BioMol) and were incubated for 0, 3, 6, 9, or 12 hours before samples were harvested into liquid nitrogen. To analyze the BR sensitivity, seeds were germinated, and seedlings were grown on ½ MS medium containing varying concentration of brassinolide (BL, Wako Chemical) for 7 days under a long-day (16-h-light/8-h-dark) photoperiodic growth condition. The seedlings were photographed, and their root length were measured digitally by ImageJ (https://imagej.nih.gov/ij/). Source data of the measurement are provided as a Source Data file. To study the BR-induced BES1 dephosphorylation, 10-day-old Arabidopsis seedlings were carefully transferred into liquid ½ MS medium supplemented with or without 1 μM BL, incubated for 2 hours, and subsequently harvested into liquid nitrogen. To study protein degradation, 10-day-old Arabidopsis seedlings were carefully transferred into liquid ½ MS medium containing 180 μM CHX (Sigma), incubated for varying durations, and harvested into liquid nitrogen for protein extraction. To study the impact of *ebs7* and *pawh1 pawh2* mutations on protein stability, 10-day-old Arabidopsis seedlings were pretreated with 5 μg/mL TM in liquid ½ MS medium for 12 hours before being treated with 180 μM CHX or 80 μM MG132 (Sigma).

**Generation of transgene constructs and transgenic plants.** A ~4.1-kb genomic fragment for each *PAWH* gene was amplified from genomic DNAs of the wild-type Arabidopsis Col-0 seedlings using the *gPAWH1* and *gPAWH2* primer sets (see Supplementary Table 1) and cloned into *pCambia1300* (https://cambia.org/welcome-to-cambialabs/) to generate *pCambia1300-gPAWH1* and *pCambia1300-gPAWH2* transgenes. The *p35S::Hrd1a-GFP* transgene was created by cloning a 1,479-bp *Hrd1a* cDNA fragment amplified from the 1st strand cDNAs converted from total RNAs of the wild-type Arabidopsis seedlings using the *Hrd1a-GFP* primer set (Supplementary Table 1) into the *pCambia1300-p35S::C-GFP* vector. To create *p35S::HA-EBS7* or *p35S::MYC-EBS7* transgene, an 876-bp *EBS7* cDNA fragment was amplified from the same Arabidopsis 1st cDNA preparation using the *HA-EBS7* and *MYC-EBS7* primer set (Supplementary Table 1) and cloned into the *pCambia1300-p35S::N-HA* or *pCambia1300-p35S::N-MYC* vector, respectively. To generate the *pEFR::EFR-FLAG* transgene, a 5.2-kb *EFR* genomic fragment (*gEFR* without its 3′-UTR sequence) was PCR amplified from the genomic DNAs of the wild-type Arabidopsis seedlings using the *gEFR-FLAG* primer set (Supplementary Table 1) and cloned into the *pCambia1300-C-FLAG* vector. The *gEBS2* construct is a genomic transgene for *EBS2*[52]. All created transgenes were fully sequenced to ensure no PCR-introduced sequence error, and were individually transformed into the *Agrobacterium tumefaciens* strain GV3101 by electroporation and the resulting Agrobacterial strains were subsequently used to transform Arabidopsis plants using the floral-dipping method[65].

**CRISPR/Cas9-mediated genome editing to create *pawh* mutants.** The CRISPR/Cas9 vectors used to create *pawh1/2-c* mutation were provided by Dr. Jian-Kang

Zhu[66]. The target sites for introducing mutations into *PAWH1* and *PAWH2* genes were selected using the web program CRISPR-PLANT (http://crispr.hzau.edu.cn/cgi-bin/CRISPR/CRISPR), and the oligonucleotides for generating guide RNAs are listed in Supplementary Table 1. The CRISPR/Cas9 constructs were introduced into the *Agrobacterium* strain GV3101 by electroporation, which were subsequently used to transform *bri1-5* mutant by the floral-dipping method[65]. T0 seeds were harvested, sterilized, and subsequently screened on ½ MS medium for T1 lines that carried the CRISPR/Cas9 constructs, which were subsequently screened for intended mutations of *PAWH1/PAWH2* genes and verified in T2 and T3 generations. The verified *pawh1-c bri1-5* and *pawh2-c bri1-5* mutants were crossed to get the *pawh1-c pawh2-c bri1-5* triple mutant.

**Expression of fusion proteins and generation of antibodies.** The 1st cDNA preparation derived from total RNAs of wild-type Arabidopsis seedlings and the *antigen-PAWH1* primer sets (Supplementary Table 1) were used to amplify a 480-bp *PAWH1* cDNA fragment corresponding to its N-terminal 160 AAs. The amplified fragment was cloned into *pGEX-4T-3* (GE Healthcare) and *pMALc-H*[67] vectors, which were subsequently transformed into BL21-competent cells. The same cDNA preparation was also used to amplify a *Hrd1a* cDNA fragment encoding the N-terminal fragment of 73 AAs using the *antigen-Hrd1a* primer sets (Supplementary Table 1), and two copies of the amplified *Hrd1a* cDNA were cloned into the *pGEX-4T-3* and *pMALc-H* vectors. The induction of GST and MBP fusion proteins and their subsequent purification using Glutathione Sepharose™ 4 Fast Flow beads (GE Healthcare) and Amylose Resin beads (New England Biolabs), respectively, were carried out following the manufacturers' recommended protocols. The purified MBP-PAWH1 and GST-Hrd1a fusion proteins were used to make custom antibody by Abmart (http://www.ab-mart.com.cn) while the purified GST-PAWH1 and MBP-Hrd1a fusion proteins were used to affinity-purify anti-PAWH and anti-Hrd1a antibodies from Abmart-generated anti-PAWH and anti-Hrd1a antisera, respectively, using an online protocol with nitrocellulose membrane (http://post.queensu.ca/~chinsang/lab-protocols/antibody-purification.html). The specificity of the purified anti-PAWH and anti-Hrd1a antibodies was analyzed by immunoblotting with total proteins extracted from 2-week-old seedlings of wild-type Arabidopsis plant and relevant mutants. Source data are provided as a Source Data file.

**Yeast two-hybrid assay.** The Clontech's yeast two-hybrid system was used to examine the interactions of PAWHs, EBS7, and Hrd1a in yeast cells (AH109). The 1st cDNA preparation described above was used to amplify cDNA fragments of the N-terminal 260 amino acids of PAWH1 (PAWH1-N260) or the N-terminal 257 amino acids of PAWH2 (PAWH2-N257) with the *Y2H-PAWH1/2* primer sets (Supplementary Table 1). The PCR-amplified *PAWH1/2* cDNA fragments were subsequently cloned into the vectors, *pGADT7* (for expressing a fusion protein of the GAL4 activation domain with a target protein) and *pGBKT7* (for producing a fusion protein of the GAL4 DNA-binding domain with a protein of interest). The *pGADT7* and *pGBKT7* constructs of EBS7-N142 and the catalytic domain of Hrd1a (Hrd1a-CD containing amino acids of 247–492) were described previously[39]. Yeast cells containing different combinations of *pGADT7* and *pGBKT7* plasmids were selected by growth on synthetic media lacking leucine and tryptophan. Several independent colonies were picked, resuspended in the liquid yeast growth medium, subsequently spotted via serial dilutions onto solid synthetic medium lacking leucine, tryptophan, and histidine for visual observation of their growth.

**RNA isolation and analysis.** Ten-day-old Arabidopsis seedlings grown on ½ MS medium supplemented with or without certain chemicals were harvested and ground in liquid nitrogen into a fine powder and their total RNAs were extracted using the RNeasy Plant Mini Kit (QIAGEN). One microgram of the purified total RNAs were treated with RNase-free DNase I (TIANGEN) and subsequently reverse transcribed into 1st strand cDNAs by the iScript™ cDNA synthesis Kit (Bio-Rad). The resulting cDNAs were used for classical PCR or quantitative real-time PCR (qPCR) analysis with gene-specific oligonucleotides listed in Supplementary Table 1. The qPCR assays were performed on the CFX96 Real-Time System (Bio-Rad) with SYBR® GREEN PCR Master Mix (Bio-Rad) following the manufacturer's instruction. Three biological replicates each with three technical repeats were conducted for each target mRNA and each sample. The *ACTIN8* cDNA was used as an internal control. The classical RT-PCR assays were carried out on a C1000 Touch™ Thermal Cycler (Bio-Rad) with 2xHieff™ PCR Master Mix (Yeasen). The PCR amplified cDNA fragments were separated by agarose gel electrophoresis and visualized by Gel Doc™ XR+ Molecular Imager (Bio-Rad) with Image Lab™ software. The *β-Tubulin* was used as an internal reference. Source data are provided as a Source Data file.

**Protein extraction and analyses.** To isolate microsomal proteins, 1 g of 10-day-old *Arabidopsis* seedlings was harvested directly into liquid $N_2$ and immediately ground in liquid nitrogen into fine powder, dissolved in a homogenization buffer [50 mM Tris-HCl (pH 8.2), 20% (v/v) glycerol, 1 mM phenylmethylsulphonyl fluoride (PMSF, Sigma), 2 mM ethylenediaminetetraacetic acid (EDTA), 1 mM DTT, 2 protease inhibitor Cocktail Tablets (Roche) per 100 mL solution], and centrifuged for 10 min at $8000 \times g$ at 4 °C. The supernatant (total proteins) was

centrifuged further at $100,000 \times g$ at 4 °C for 60 min to collect the microsomal pellet. The resulting pellet was dissolved in the homogenization buffer as the microsomal fraction. Both the total proteins and microsomal fraction were used to immunoprecipitate the GFP-tagged Hrd1a and MYC/HA-tagged EBS7 using anti-GFP mAb-agarose (D153-8, MBL) and anti-MYC/anti-HA-agarose affinity gels (Sigma) for mass spectrometric analysis of Hrd1/EBS7-associated proteins. The microsomal fraction of the wild-type seedlings was subsequently treated with solutions of 0.1 M NaCl, 0.1 M $Na_2CO_3$, 1% (v/v) Triton X-100 (Sigma) or 1% (v/v) Nonidet P-40 (Roche) at 4 °C for 4 h. After treatment, the four fractions were centrifuged at $100,000 \times g$ at 4 °C for 60 min to collect the supernatants (soluble fraction) and the pellets, which were mixed with 2x SDS buffer, boiled for 10 min, separated by 12% SDS/PAGE, and analyzed by immunoblotting.

We used 10-day-old seedlings to perform the coIP assays[39]. In brief, 10-day-old Arabidopsis seedlings were harvested, ground into fine powder in liquid $N_2$. Total proteins were extracted in the extraction buffer [50 mM Tris-HCl (pH 8.0), 150 mM NaCl, 5 mM EDTA, 0.1% (v/v) Triton X-100, 0.2% (v/v) Nonidet P-40 (Roche), 2 tablets of the protease inhibitor cocktail (Roche) per 100 mL] and the extracts were centrifuge at $14,000 \times g$ for 15 min at 4 °C. The resulting supernatants were incubated with anti-EBS7 antibody[39] followed by protein-A/G-agarose beads (Abmart) at 4 °C for 8 h. The immunoprecipitates were washed five times with the extraction buffer, resuspended in 2X SDS sample buffer, boiled at 95 °C for 10 min, and separated by SDS/PAGE. For other immunoblot assays, 10-day-old Arabidopsis seedlings treated with or without a chemical of interest were harvested into liquid $N_2$ and stored in −80 °C freezer for later analysis or immediately ground into fine powder in liquid $N_2$. The broken plant tissues were dissolved in 2x SDS sample buffer, boiled at 95 °C for 10 min, and centrifuged for 10 min at the top speed in an Eppendorf 5424 centrifuge. The resulting supernatants were immediately used for immunoblotting analysis or incubated with or without 1000 U Endo-Hf in 1x G5 buffer (New England Biolabs) at 37 °C for 1.5 h. The treated/non-treated total proteins were separated by 7, 10, or 12% SDS-PAGE. For all the immunoblotting performed throughout this study, after transfer from SDS-PAGE gels to PVDF membranes (Bio-Rad), proteins were analyzed by immunoblot with antibodies against BES1[50], BRI1[50], BiP (at-95, Santa Cruz Biotechnology), maize-CRT[68], PDI (Rose Biotechnology Inc), EBS5[36], EBS6[37], EBS7[39], PAWHs (this study), Hrd1a (this study), GFP [living color® A.v.monoclonal antibody (JL-8), 632381, Takara], or HA (H9658, Sigma). The signals on those immunoblots were detected with horseradish peroxidase-conjugated secondary antibodies, ECL Select™ Western Blotting Detection Reagent (GE Healthcare), and Medical X-ray Processor (Kodak). The resulting signals on X-ray films were scanned and digitalized. Source data are provided as a Source Data file.

**Sucrose density-gradient centrifugation**. Sixteen grams of 10-day-old Arabidopsis seedlings were ground in liquid $N_2$ into a fine powder and immediately extracted by the homogenization buffer (see above) at 4 °C. The protein extracts were first filtered through Miracloth (CalBiochem) to remove insoluble plant debris and subsequently centrifuged at $5000 \times g$ for 5 min at 4 °C to remove cellular debris and organelles. The supernatant was centrifuged at $100,000 \times g$ for 45 min to pellet the microsomes, which was resuspended in 1 mL resuspension buffer [25 mM Tris-HCl (pH 7.5), 10% (w/v) sucrose, 1 mM PMSF, 2 mM EDTA, 1 mM DTT, 2 protease inhibitor Cocktail Tablets (Roche) per 100 mL]. The microsome resuspension was loaded onto the top of a 11-mL 20–50% (w/w) sucrose gradient in 10 mM Tris-HCl (pH 7.5), 2 mM EDTA, 1 mM DTT, 0.1 mM PMSF, and centrifuged at $100,000 \times g$ for 16 h. After centrifugation, 1 mL fractions (12 fractions) were hand collected, and 50 μL protein sample for each fraction was mixed with 2x SDS buffer, boiled for 10 min, separated by 10 and 12% SDS/PAGE, and analyzed by immunoblotting. For the $Mg^{2+}$-plus experiments, 5 mM $MgCl_2$ was added to the buffers of homogenization, resuspension, and ultracentrifugation. Source data are provided as a Source Data file.

**Confocal analysis of PAWH1/2 fusion proteins**. To generate p35S::GFP-PAWH1 and p35S::GFP-PAWH2 transgenes for visualization of PAWH1/2 subcellular localization patterns, an 858-bp PAWH1 cDNA fragment and an 849-bp PAWH2 cDNA fragment were amplified from the 1st cDNA preparation of the wild-type Arabidopsis seedlings using the GFP-PAWH1 and GFP-PAWH2 primer sets (Supplementary Table 1) and subsequently cloned into the pCambia1300-p35S::N-GFP vector. The plasmid p35S::RFP-HDEL was the same as used in the EBS7 study[39]. To analyze the protein interaction using the bimolecular fluorescence complementation (BiFC) assay, an 858-bp PAWH1 cDNA and an 849-bp PAWH2 cDNA fragment amplified from the 1st strand cDNAs of wild-type Arabidopsis seedlings using the NER-PAWH1 or NER-PAWH2 primer set (see Supplementary Table 1) was cloned into the pVYNER vector[69]. The same 1st cDNA preparation was also used to amplify the full-length (876-bp) EBS7 cDNA using the CER-EBS7 primer set and the full-length (1479 bp) Hrd1a cDNA using the CER-Hrd1a primer set (Supplementary Table 1), which were subsequently cloned into the pVYCER and pVYCE vectors[69]. These plasmids and the corresponding empty vectors, after verifying no PCR-introduced error, were individually transformed into the Agrobacterium tumefaciens strain GV3101. The p35S::RFP-HDEL-carrying GV3101 cells were mixed with the p35S::GFP-PAWH1 or p35S::GFP-PAWH2-transformed GV3101 strain and co-transformed into leaves of 3-week-old tobacco (Nicotiana benthamiana) plants by the agro-infiltration

method[70]. Similarly, a mixture of two Agrobacterium strains, one carrying a pVYNER plasmid and the other containing a pVYCER or pVYCE plasmid, was used to infiltrate young leaves of 3-week-old tobacco. Two days after infiltration, the transformed tobacco leaves were examined by confocal microscopy on a Leica SP8 (with LAS AF software, Leica Microsystems) for the subcellular localization patters of GFP-tagged PAWH1/2 and the ER-localized RFP-HDEL or the reconstituted YFP signal. GFP, YFP, and RFP were excited by using 488-, 514-, and 542-nm laser light, respectively.

**Liquid chromatography-tandem mass spectrometry**. Immunoprecipitated proteins were eluted from antibody-conjugated beads with 0.2 M Glycine (pH 2.5), dried, and solubilized in 8 M urea containing 50 mM iodoacetamide. The alkylated proteins were digested by sequencing-grade trypsin (Promega) in 25 mM $NH_4HCO_3$ at an enzyme:substrate ratio of 1:100 (w/w) in a 37 °C shaking incubator for 18 h. The tryptic peptides were collected after centrifugation at $13,523 \times g$ and freeze-dried with a refrigerated CentriVap concentrator (Labconco). Protein samples were reconstituted in 0.1% formic acid (FA) and analyzed by online nanoAcquity ultraperformance LC (Waters) coupled with an Orbitrap Fusion Tribrid mass spectrometer (Thermo Fisher Scientific). Nanospray was controlled by a PicoView Nanospray Source (PV550; New Objective) at a spray voltage of 1.9 kV. The peptides were trapped by a 2G-V/MT Trap symmetry C18 column (5 μm particles, 180 μm inner diameter × 20 mm length) at a flow rate of 3 μL min$^{-1}$ for 3 min, and separated on a BEH130 C18 analytical column (1.7 μm particles, 100 μm inner diameter × 250 mm length) at a flow rate of 250 nL min$^{-1}$. Peptides were eluted using mobile phases consisting of solvent A (0.1% FA in water) and solvent B (0.1% FA in acetonitrile) through a linear gradient, and then 85% of solvent B at a duration of 90 min. Data-dependent MS/MS acquisition was performed following a full MS survey scan by Orbitrap at a resolution of 70,000 over the m/z range of 350–1800, and MS/MS measurements of the top 20 most intense precursor ions. The MS raw data from the LC-MS/MS analyses were separately converted to mascot generic format (MGF) files using the Proteome Discoverer 1.4 software (Thermo Fisher Scientific) and then searched against in-house databases using Mascot Daemon 2.4 (MatrixSciences). The search parameter for tryptic digestion was restricted to a maximum of two missed cleavages of proteins. Cysteine carbamidomethylation was designated as a fixed modification. Mass tolerances were set up to 20 ppm for the Orbitrap-MS ions, and 0.2 or 1 Da for ion-trap MS/MS fragment ions.

**Reporting summary**. Further information on research design is available in the Nature Research Reporting Summary linked to this article.

## Data availability

The mass spectrometry proteomic data have been deposited to the ProteomeXchange Consortium via the PRIDE partner repository (https://www.ebi.ac.uk/pride/archive/) with the dataset identifier PXD013400. The source data underlying Figs. 1d, 2b, c, 3d–f, 4a, 5a–c, and 6a–d and Supplementary Figs. 6, 9b, c, 11b–d, 12, 13b, 14b, c, e, f, 15d–f, 16d, 17–19, 20b, and 21 are provided by a Source Data file. The source data for generating Supplementary Figs. 7 and 8 were obtained from http://bar.utoronto.ca/efp_arabidopsis (using At4g17420 and At5g47420 as queries) and atted.jp (using At5g47420 as query), respectively. All other data are available from the corresponding authors upon a reasonable request.

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

## Acknowledgements

We are grateful to the Core Facilities at Shanghai Center for Plant Stress biology for their excellent technical supports, the Arabidopsis Biological Resource Center at Ohio State University for providing seeds of the T-DNA insertional *pawh1* and *pawh2* mutants, Dr. Frans Tax for *bri1-5* seeds, Dr. Yanhai Yin for anti-BES1 antibody, and Dr. Jian-Kang Zhu for the CRISPR/Cas9 vectors. We thank other members of the J.L. laboratory for stimulating discussions. This work was partially supported by grants from National Natural Science Foundation of China (NSFC31730019 to J.L. and NSFC31600996 to L.Liu.), a grant from the Chinese Academy of Sciences (2012CSP004 to J.L.), and research funds from Shanghai Center for Plant Stress Biology and South China Agricultural University.

## Author contributions

J.L. conceived the research plans, J.L., L.Liu., and W.X. supervised the project, suggested experiments, analyzed results, and wrote the manuscript with support from J.M. and J.Z.; C.Z. initiated the project and carried out the immunoprecipitation-mass spectrometry experiments, L.Lin. performed a majority of the experiments, analyzed data, and prepared figures with technical assistance from Y.C, Y.W., D.W, X.L., and M.W.

## Additional information

**Competing interests:** The authors declare no competing interests.

