## [Peer Review File · Nature Communications]

Reviewers' comments:

Reviewer #1 (Remarks to the Author):

This very interesting study provides novel, perhaps unexpected, information on players and interactions in plant ERAD. The two newly identified proteins, PAWH1 and PAWH2, belong to a larger protein family. To my knowledge members of this family had not been previously implicated in ERAD. The authors provide extensive biochemical and genetic data to support their conclusions. The approaches and data are in general of high quality, however the two following experiments have problems that need to be solved:

1. Supplementary Fig. 18. This information is used at several points in the manuscript, however statistical analysis should be used to determine the significance of the differences observed. Some of the cited differences seem to be very marginal and possibly statistically insignificant. Without additional statistical analysis, it is questionable to state, for example, that: "By contrast, the ebs6 mutation had little effect on the protein abundance of EBS7, PAWHs, or Hrd1a, whereas the ebs5 mutation actually increased the protein levels of these proteins (Fig. 5a) likely due to increased abundance of the correspondent transcripts (Supplementary Fig. 18)." Or that "our quantitative real-time PCR (qPCR) analyses showed that the pawh1 pawh2 double mutation slightly elevated the levels of the EBS7 and Hrd1a transcripts (Supplementary Fig. 18)."
2. I am puzzled by the sucrose gradient fractionation in Fig. 2. Especially without Mg⁺⁺, the density of PM microsomes should be much higher than that of ER microsomes (see for example FEBS Letters 579 (2005) 5814–5820). Which are the densities of the fractions? Also the two very distinct ER peaks are unusual. Something seems to be wrong in those gradients.

To help readers in understanding the model deriving from this study for the interactions of ERAD proteins, I also strongly suggest the authors to summarize their findings in a cartoon, maybe building on the one they had prepared as Figure 3 in *Front. Plant Sci.* 5:162.

Additional points

- a. In the Abstract, line 5, I would omit "very" from the statement "however, our knowledge of this complex is very limited.", and I would do the same in the similar sentence on page 4. Our knowledge on that is limited, but not very limited.
- b. I do not understand what Supplementary Fig. 2 represents. According to Fig. 1 there are 9 proteins common to the three immunoprecipitations and absent in WT, however Supplementary Figs. 2b, c and d show 4, 7 and 7 proteins only, respectively.
- c. Legend of Supplementary Fig. 6. line 5: I think that "while lines" should be "white line".
- d. Legend of Supplementary Fig. 17. Is the construct EFR-FLAG or EFR-GFP?
- e. Page 13. "Interestingly, the ebs7-1 mutation also significantly reduced the protein abundance of the two PAWHs (Fig. 5a)." The figure shows a reduction of PAWH1 but not PAWH2.
- f. Fig. 5a. What does the asterisk indicate?
- g. Fig 6. What do the asterisk, star and small rectangle indicate?
- h. Page 17. "the CHX treatment caused a gradual reduction of EBS7 but very rapid disappearance of the TM-induced Hrd1a (Fig. 6c)". I do not see a rapid disappearance of Hrd1a.

Reviewer #2 (Remarks to the Author):

Nature Communications manuscript NCOMMS-18-5243735

Title: A crucial role of two Arabidopsis homologs of the yeast AIM24 protein in endoplasmic reticulum-associated degradation.

In this original contribution, authors have identified and characterized components of Hrd1-ERAD

machinery thanks to an IP-MS strategy in *Arabidopsis thaliana*. They especially identified PAWH1 and 2 and confirmed the interaction with EBS7 using an IP followed by immunoblotting. Moreover, they added some details about interaction between the three partners Hrd1/PAWH and EBS7, shown that PAWH1/2 are anchored to the ER membrane then demonstrated that these two proteins are crucial components of Hrd1-ERAD complex. The authors afterwards highlighted that the pawh1 pawh2 double mutation constitutively activates the UPR pathway and compromises stress tolerance. Moreover, the double mutation reduces the protein level of other Hrd1-ERAD complex components and inhibits the degradation of several ERAD substrates.

The article is an in depth study of Hrd1-ERAD complex composition and regulation in plants. The manuscript is clearly written, interpretations and conclusions are overall justified by the data and reasoning easy to follow.

Consequently I recommend the manuscript for publication in Nature Communications.

I just have some questions and want to suggest some points:

Since the authors have the antibodies raised against EBS7, Hrd1 and PAWH available, have they try to perform IP-MS from endogenous proteins?

In the case you performed co-IP from tagged proteins, better control would have been obtain with GFP/HA and MYC IP-MS alone. Indeed, the negative control described by the authors allowed to eliminate proteins linked to beads or antibodies but not partners linked to tags.

Conclusions from bri1-9 mutant are not always clear enough. I suggest a more detailed description of this cell line in the introduction in order to better follow conclusions from experiments with this mutant.

Sometimes figures are not enough described: for instance Fig.5b and c, where the authors directly concluded without description of all the results.

In return the conclusion lacks after fig.6 description.

I think the title could be ameliorated with mention of PAWH as ERAD components instead of AIM24 yeast homologs.

Minor comment:

Please add space between units and values in materials and methods (for instance 0.1M; 8h; 12%; 1Da...).

There is a problem with supplementary fig.17: has the experiment been performed with GFP or with FLAG tagged EFR protein? Legend, figure and material and methods are not in agreement. Please take note of these remarks.

Reviewer #3 (Remarks to the Author):

In this article, authors sought novel members of ERAD machinery in *Arabidopsis*. To do so, they took proteomic approach, namely they identified proteins coimmunoprecipitated with Hrd1a, a ubiquitin E3 ligase, and EBS7 that regulates the stability of Hrd1a. This experiment discovered two PAWH proteins that have similarity with AIM24 domain. Subsequent three experiments, Y2H, BiFC and CoIP, showed that PAWHs interact with EBS7 but not with Hrd1a directly. Following experiments revealed induction of PAWHs by ER stress like as other ERAD components. PAWHs were found to be localized on the ER membrane from observation of localization of GFP fusion proteins and subcellular fractionation experiments, although they do not contain signal peptide or ER retention signal. Disruption of PAWH genes inhibited degradation of BRI1 and other ERAD substrate such as EFR. Since UPR genes that are induced by ER stress such as tunicamycin, are expressed in the mutant of PAWHs without and tunicamycin, constitutive UPR was though to occur in the mutant. In addition, the mutant is more sensitive to DTT and NaCl than wild type indicating reduction of ER-QC ability. In ebs7-1 mutant, PAWH proteins are unstable as well as Hrd1a. Hrd1a is also unstable in PAWHs.

From above experiments PAWH1 and PAWH2 are considered to be novel components in Hrd1 ERAD complex. Although PAWHs are supposed not to interact with Hrd1, they seem stabilize

Hard1 on EBS7 dependent manner. The mechanism of stabilization of Hard1 through EBS7 by PAWH1 and PAWH2 remains unclear. In discussion, authors propose model using crystal structures of bacterial AIM24 domain that PAWHs also contain.

It was pleasant to read this manuscript. Experiments were designed smartly, most of results were clear and interpretations of result are logical. It is interesting that plants contain ERAD components different from yeast and animals. It is worth to discuss correlation between the difference of molecular composition in ERAD components and ERAD functions, if there are any.

In addition, authors are recommended to reconsider following minor issues.

1. Is "homologs of the yeast AIM24" in title suitable description? PAWHs contains AIM24 domain, but are they homologs of AIM24? More suitable title is recommended.
2. In Fig. 1B, authors identified four proteins in addition to PAWHs, however authors mentioned nothing. Please just describe something very briefly.
3. In Fig3d, BL is supposed be BR, isn't it? In Fig. 3e, signals in pawh1 pawh2 bri1-9 are stronger than those in WT. Is there any reason? Check legend of Fig.3 carefully.
4. In Fig 4b and c, the pawh1 pawh2 mutant is more sensitive DTT and NaCl. The result of DTT makes sense since DTT is a popular UPR inducer, while why NaCl? Any relation between UPR and NaCl?
5. Please add explanation of a star and an asterisk in Fig. 6, even they are just nonspecific signal or something.

Nozomu Koizumi

Dear Dr. Pattison:

Thank you very much for handling our manuscript and obtained critical comments and helpful suggestions from three anonymous reviewers. We were quite encouraged by the supportive comments from all three anonymous reviewers. Based on their constructive comments and the manuscript checklist, we revised our manuscript. Please see below for our point-to-point responses to reviewers' criticisms/suggestions. I also included a summary of what we have done to comply with the requirement of Nature Communication manuscripts.

Reviewer #1

(1) Supplementary Fig. 18. This information is used at several points in the manuscript, however statistical analysis should be used to determine the significance of the differences observed. Some of the cited differences seem to be very marginal and possibly statistically insignificant. Without additional statistical analysis, it is questionable to state, for example, that: "By contrast, the ebs6 mutation had little effect on the protein abundance of EBS7, PAWHs, or Hrd1a, whereas the ebs5 mutation actually increased the protein levels of these proteins (Fig. 5a) likely due to increased abundance of the correspondent transcripts (Supplementary Fig. 18)." Or that "our quantitative real-time PCR (qPCR) analyses showed that the pawh1 pawh2 double mutation slightly elevated the levels of the EBS7 and Hrd1a transcripts (Supplementary Fig. 18)."

Response: We agree with this reviewer that a statistical analysis should be performed for our qPCR results. We have therefore performed statistical analysis of our qPCR data using the Graphpad Prism 7 to determine the significance of the differences of analyzed Arabidopsis genes in a given mutant in comparison to the *bri1-9*. We changed the Supplementary Figure 18 with the new bar graph containing the statistical results. We modified the conclusion and discussion that related to the results of the Supplementary Fig. 18.

(2) I am puzzled by the sucrose gradient fractionation in Fig. 2. Especially without Mg⁺⁺, the density of PM microsomes should be much higher than that of ER microsomes (see for example FEBS Letters 579 (2005) 5814–5820). Which are the densities of the fractions? Also the two very distinct ER peaks are unusual. Something seems to be wrong in those gradients.

Response: We appreciate the criticism of this reviewer and carefully repeated the sucrose gradient ultracentrifugation experiment by following the protocol suggested by the reviewer. The results of the experiment were included in the revised Fig. 2. As shown in Fig. 2b, the density of the PM microsome was higher than that of ER microsomes, and there was only one peak of the ER microsomes that was shifted to a higher density in the presence of Mg²⁺.

(3) To help readers in understanding the model deriving from this study for the interactions of ERAD proteins, I also strongly suggest the authors to summarize their findings in a cartoon, maybe building on the one they had prepared as Figure 3 in Front. Plant Sci. 5:162.

Response: Following this reviewer's suggestion, we created a model of the Arabidopsis Hrd1-containing ERAD complex, which was presented in a new figure, Fig. 7, along with a

summary statement in the “Discussion” section and its figure legend. We are hoping that the model will be helpful to appreciate the significance of our discovery.

Addition points

(a) *In the Abstract, line 5, I would omit “very” from the statement “however, our knowledge of this complex is very limited.”, and I would do the same in the similar sentence on page 4. Our knowledge on that is limited, but not very limited.*

Response: Following this review’s suggestion, we deleted the word “very”.

(b) *I do not understand what Supplementary Fig. 2 represents. According to Fig. 1 there are 9 proteins common to the three immunoprecipitations and absent in WT, however Supplementary Figs. 2b, c and d show 4, 7 and 7 proteins only, respectively.*

Response: We apologize for the confusion. The 9 common proteins were discovered by 4 independent IP-MS experiments that include three using total protein extracts and one using the microsomal preparation. The three total protein-IP-MS experiments also identified Hrd1b and EBS6 that were missed by the microsome-IP-MS assay, which is likely caused by its low recovery of immunoprecipitated proteins. The 4 lists of proteins presented in Supplementary Fig. 2 only list known components of the Arabidopsis ERAD complex plus the two PAWH proteins but not the 4 other common proteins that were indicated in Fig. 2b.

(c) Legend of Supplementary Fig. 6. line 5: I think that “while lines” should be “white line”.

Response: Thanks for identifying the typo. We changed “while” to “white”.

(d) Legend of Supplementary Fig. 17. Is the construct EFR-FLAG or EFR-GFP?

Response: We apologize for the mistake. The construct of Supplementary Fig. 17 was EFR-FLAG, and we made necessary changes to correct the mistake.

(e) Page 13. “Interestingly, the ebs7-1 mutation also significantly reduced the protein abundance of the two PAWHs (Fig. 5a).” The figure shows a reduction of PAWH1 but not PAWH2.

Response: We apologize for the mistakes. The band with asterisk was an unspecific band that cross-reacts with the anti-PAWH antibody, and the higher band was PAWHs. The two PAWH bands were not separated well in this immunoblot analysis.

(f) Fig. 5a. What does the asterisk indicate?

Response: The asterisk indicates a cross-reacting band.

(g) Fig 6. What do the asterisk, star and small rectangle indicate?

Response: Sorry about the errors that we made. We removed the small rectangle, added explanation for the asterisks and stars in the legend, and added a black arrow that indicates the position of Hrd1a.

(f) Page 17. “the CHX treatment caused a gradual reduction of EBS7 but very rapid disappearance of the TM-induced Hrd1a (Fig. 6c)”. I do not see a rapid disappearance of

Hrd1a.

Response: We apologize that we did not explain the meaning of the asterisk and rectangle in Fig.6c. The band of Hrd1a in Fig. 6c was indicated by red arrow.

Reviewer #2

(1) Since the authors have the antibodies raised against EBS7, Hrd1 and PAWH available, have they try to perform IP-MS from endogenous proteins?

In the case you performed co-IP from tagged proteins, better control would have been obtain with GFP/HA and MYC IP-MS alone. Indeed, the negative control described by the authors allowed to eliminate proteins linked to beads or antibodies but not partners linked to tags.

Response: We thank the reviewer for asking the question. Among the antibodies that we have, only the anti-EBS7 antibody can be used for immunoprecipitation experiments (in other words, only the anti-EBS7 can recognize the endogenous EBS7 with its native conformation and/or with its interacting partners). Although we did not perform an IP-MS experiment with the anti-EBS7 antibody and non-transgenic wild-type plants, we did use the anti-EBS7 antibody to perform two co-IP experiments that were presented in Fig. 1d and Supplementary Fig. 21, which confirmed the EBS7-PAWH1/2-Hrd1 association in wild-type plants but not in the null *ebs7-3* mutant plants.

(2) *Conclusions from bri1-9 mutant are not always clear enough. I suggest a more detailed description of this cell line in the introduction in order to better follow conclusions from experiments with this mutant.*

Response: Thanks for the suggestion. We added two sentences in the 1st paragraph of the “Result” section to explain the relationship between the ER-retained mutant BR receptors *bri1-5* and *bri1-9* and their corresponding dwarf mutants *bri1-5* and *bri1-9*. We also explained why loss-of-function mutations in ERAD components not only inhibit degradation of the two mutant BR receptors but also suppress the growth phenotypes of the corresponding dwarf mutants. I am hoping that the added explanation will be helpful to understand the effects of the *pawh1 pawh2* double mutation and an overexpression transgene of EBS2 (one of the three Arabidopsis homologs of calreticulin) that was previously shown to retain *bri1-9* in the ER.

(3) Sometimes figures are not enough described: for instance, Fig.5b and c, where the authors directly concluded without description of all the results. In return the conclusion lacks after fig.6 description.

Response: Thanks for the comments. Following this reviewer’s suggestion, we added the result description of Fig. 5b and 5c and the conclusion for the results presented in Fig. 6.

(4) I think the title could be ameliorated with mention of PAWH as ERAD components instead of AIM24 yeast homologs.

Response: We appreciate the suggestion, which was also made by the reviewer #3. Following their suggestions and the journal’s requirement, we changed the title of the revised manuscript to “PAWH1 and PAWH2 are plant-specific components of an Arabidopsis endoplasmic reticulum-associated degradation complex”.

(5) Please add space between units and values in materials and methods (for instance 0.1M; 8h; 12%; 1Da...).

There is a problem with supplementary fig.17: has the experiment been performed with GFP or with FLAG tagged EFR protein? Legend, figure and material and methods are not in agreement.

Response: Thanks for the comments. We added space between units and values in the “Materials and Methods” section. We also corrected the errors on the EFR-FLAG transgene..

Reviewer #3

(1) *Is “homologs of the yeast AIM24” in title suitable description? PAWHs contains AIM24 domain, but are they homologs of AIM24? More suitable title is recommended.*

Response: Thanks for the suggestion. We changed the title of our revised manuscript to “PAWH1 and PAWH2 are plant-specific components of an Arabidopsis endoplasmic reticulum-associated degradation complex”.

(2) *In Fig. 1B, authors identified four proteins in addition to PAWHs, however authors mentioned nothing. Please just describe something very briefly.*

Response: Thanks for the suggestion. We added two sentences (the last two sentences of the 1st paragraph on page 6) explaining the identity of the 4 other common proteins that were identified by 4 IP-MS experiments.

(3) *In Fig3d, BL is supposed be BR, isn't it? In Fig. 3e, signals in pawh1 pawh2 bri1-9 are stronger than those in WT. Is there any reason? Check legend of Fig.3 carefully.*

Response: Thanks for asking the questions. BL is the abbreviation for brassinolide, which is the most active member of the BR family. We added this explanation in the legends of main figures and supplementary figures. Yes, the bri1-9 signal in the *pawh1 pawh2* double mutant was stronger than the signal of wild-type BRI1. This is likely caused by strong inhibition of bri1-9 degradation by the *pawh1 pawh2* double mutation while the wild-type BRI1 protein undergoes its phosphorylation-dependent ubiquitin-mediated degradation. We added the explanation with a cited reference to the main text.

(4) *In Fig 4b and c, the pawh1 pawh2 mutant is more sensitive DTT and NaCl. The result of DTT makes sense since DTT is a popular UPR inducer, while why NaCl? Any relation between UPR and NaCl?*

Response: Thanks for asking. Several published works have shown that salt stress activates the unfolded protein response pathway in Arabidopsis, which involving several membrane anchored bZIP transcriptional factors including bZIP17, bZIP28, and bZIP60. We changed the sentence “The detected enhancement of UPR prompted us to test if the *pawh1 pawh2* double mutation affects the plant stress tolerance” to “The detected enhancement of UPR prompted us to test if the *pawh1 pawh2* double mutation affects the plant stress tolerance, which is known to involve the UPR pathway⁵⁵” citing a recent review article (Howell, 2013, Annu Rev Plant Biol 64:477-499).

(5) Please add explanation of a star and an asterisk in Fig. 6, even they are just nonspecific signal or something.

Response: Thanks for the comment. We corrected the mistakes and added the explanation for the asterisk and the star. We also added a black arrow to indicate the position of Hrd1a.

In addition to the above point-to-point responses to address concerns/suggestions from the three anonymous reviewers, we also revised the manuscript to comply with the requirement of Nature Communication manuscripts:

We deleted some references so that the total number of the cited references for the main text is 70. We also shortened the abstract so that the total words is 150. We also shortened the main text of the revised manuscript so that the total number of words for Introduction, Results, and Discussion is within the 5,000 word-limit. We also added DNA/protein markers to the DNA gel images/immunoblot images. We also modified the subheads of the “Results” and “Discussion” sections so that each subhead contain <60 characters (including spaces).

We also replaced the gel images of the RT-PCR analysis (in Supplementary Fig. 11c) that revealed the null nature of the *pawh1 pawh2* double mutant, because we could not locate the digital files of the original RT-PCR gels. We isolated the total RNAs from seedlings of wild-type and the *pawh1 pawh2* mutant, performed semi-quantitative RT-PCR analysis, ran the resulting RT-PCR productions on gels, and obtained similar results. The new results were used to generate the revised Supplementary Fig. 11c and the full-scan images of the RT-PCR gels were included the Source Data file.

As required, we also submitted our proteomic raw data to the ProteomeXchange consortium via the PRIDE partner repository with the dataset identifier PXD013400. These IP-MS results have provided a solid experimental foundation to launch the research project that investigates the physiological and biochemical functions of the two PAWH protein in the Arabidopsis Hrd1-containing ERAD complex. We also generated a Source Data file (LinSourcedata.exe) that include the raw data that were used to generate the following figures: Figs. 1d, 2b,c, 3d,e,f, 4a, 5a,b,c, 6a,b,c,d plus Supplementary Figs. 6, 9b,c, 11b,c,d, 12, 13b, 14b,c,e,f, 15d,e,f, 16d, 17, 18, 19, 20b, 21. A data availability statement was inserted between “Methods” and “Figure Legends”.

We are hoping that the revision is satisfactory to you and the three anonymous reviewers. Please feel free to contact me if additional modifications are needed in order to meet the Journal’s requirement. We are looking forward to hearing back from you soon.

Sincerely yours,

Jianming Li

REVIEWERS' COMMENTS:

Reviewer #1 (Remarks to the Author):

The authors have considered my observations and suggestions, performing new experimental work and statistical analysis. The statistical analysis of the data in Supplementary Fig. 18 is in general properly interpreted, with the exception of the following statements, which do not change the conclusions of the manuscript but nevertheless need to be revised because they are not correct. Page 10. "which could be caused by nonsignificant and significant impact of the ebs6 and ebs5 mutations, respectively, on the transcript abundance of EBS7, PAWHs, and Hrd1a (Supplementary Fig. 18).". This must be changed, because the impact of ebs5 on Hrd1a is not statistically significant. Similarly, on page 11, line 5 "...analyses showed that the pawh1 pawh2 double mutation slightly elevated the levels of the EBS7 and Hrd1a transcripts (Supplementary Fig. 18)." Is not a correct statement, since Supplementary Fig. 18 indicates that the difference of Hrd1a between WT and the pawh1 pawh2 double mutation is not statistically significant.

Minor points

1. Abstract, line 5: change "...and PAWH2 that share..." To "...and PAWH2, which share..."
2. Page 3, fourth line from the end of the Introduction. PAWH1 and 2 may be more specifically defined as paralogs, since they originated in the same species.
3. Supplementary Fig. 7. Are you sure that the figures comes from Winter et al, 2007?
4. Supplementary Fig. 11. Describe in the legend what the asterisks indicate.
5. Page 15, line 8. "Supplementary Fig. 9" cited here is actually Supplementary Fig. 10.

Reviewer #2 (Remarks to the Author):

In the revised version the authors have carefully considered comments on the original manuscript and corrected. I still have one minor concern:

- The revised version does not mention the Fig. 5c in the text. Please correct it.

Reviewer #3 (Remarks to the Author):

The current manuscript is properly revised according to reviewers' comments including mine. Especially, Figure 7 is very helpful for understanding of readers.

Nozomu Koizumi

I want to thank the three reviewers again for their careful reading of the revised manuscript. Based on their suggestions, I revised the manuscript again and made the suggested changes. Please see below for my point-to-point responses to the reviewers' comments.

Reviewer #1:

The authors have considered my observations and suggestions, performing new experimental work and statistical analysis. The statistical analysis of the data in Supplementary Fig. 18 is in general properly interpreted, with the exception of the following statements, which do not change the conclusions of the manuscript but nevertheless need to be revised because they are not correct. Page 10. "which could be caused by nonsignificant and significant impact of the ebs6 and ebs5 mutations, respectively, on the transcript abundance of EBS7, PAWHs, and Hrd1a (Supplementary Fig. 18)." This must be changed, because the impact of ebs5 on Hrd1a is not statistically significant. Similarly, on page 11, line 5 "...analyses showed that the pawh1 pawh2 double mutation slightly elevated the levels of the EBS7 and Hrd1a transcripts (Supplementary Fig. 18)." Is not a correct statement, since Supplementary Fig. 18 indicates that the difference of Hrd1a between WT and the pawh1 pawh2 double mutation is not statistically significant.

RESPONSE: Thanks for the comments. We made the following changes:

The two words "and Hrd1a" were deleted from the end of the sentence "which could be caused by nonsignificant and significant impact of the ebs6 and ebs5 mutations, respectively, on the transcript abundance of EBS7, PAWHs, and Hrd1a (Supplementary Fig. 18)." The new sentence is "which could be caused by nonsignificant and significant impact of the ebs6 and ebs5 mutations, respectively, on the transcript abundance of EBS7 and PAWHs (Supplementary Fig. 18)." Similarly, we also changed the sentence on Page 11 "...analyses showed that the pawh1 pawh2 double mutation slightly elevated the levels of the EBS7 and Hrd1a transcripts (Supplementary Fig. 18)." to "...analyses showed that the pawh1 pawh2 double mutation slightly elevated the levels of the EBS7 mRNA but had a marginal impact on the Hrd1a transcript abundance (Supplementary Fig. 18)."

Minor points:

1. Abstract, line 5: change "...and PAWH2 that share...." To "...and PAWH2, which share...."
2. Page 3, fourth line from the end of the Introduction. PAWH1 and 2 may be more specifically defined as paralogs, since they originated in the same species.
3. Supplementary Fig. 7. Are you sure that the figures comes from Winter et al, 2007?
4. Supplementary Fig. 11. Describe in the legend what the asterisks indicate.
5. Page 15, line 8. "Supplementary Fig. 9" cited here is actually Supplementary Fig. 10.

RESPONSE: We really appreciate this reviewer's comments/suggestions:

1. We made the suggested change.
2. We changed the sentence "Our subsequent biochemical and genetic studies demonstrated that two homologous Arabidopsis proteins..." to "Our subsequent biochemical and genetic studies demonstrated that two paralogous Arabidopsis proteins...".
3. We deleted the last sentence "Reprinted with permission from a previously published study³", and added the reference to the web link http://bar.utoronto.ca/efp_Arabidopsis.
4. We added "The asterisks indicate two non-specific cross-reacting bands." to the legend of **Supplementary Fig. 11**.
5. We corrected the mistake.

Reviewer #2.

-The revised version does not mention the Fig. 5c in the text. Please correct it.

RESPONSE: Thanks for spotting the mistake. We made the suggested correction. We added "(Fig. 5c)" to the end of the sentence near the bottom of page 10. The new sentence is "...whereas the same MG132 treatment actually increased the PAWH1/PAWH2 protein levels in the *ebs7-1* mutant (**Fig. 5c**)".